# MOSAIC: Multiple Observers Spotting AI Content, a Robust Approach to Machine-Generated Text Detection

## Abstract

The dissemination of Large Language Models (LLMs), trained at scale, and endowed with powerful text-generating abilities has vastly increased the threats posed by generative AI technologies by reducing the cost of producing harmful, toxic, faked or forged content. In response, various proposals have been made to automatically discriminate artificially generated from human-written texts, typically framing the problem as a classification problem. Most approaches evaluate an input document by a well-chosen detector LLM, assuming that low-perplexity scores reliably signal machine-made content. As using one single detector can induce brittleness of performance, we instead consider several and derive a new, theoretically grounded approach to combine their respective strengths. Our experiments, using a variety of generator LLMs, suggest that our method effectively leads to robust detection performances.

## 1 Introduction

Large Language Models (LLMs) have greatly improved the fluency and diversity of machine-generated texts. The release of ChatGPT and GPT-4 by OpenAI has sparked global discussions regarding the effective use of AI-based writing assistants. This progress has also introduced considerable threats such as fake news generation Zellers et al. (2019), and the potential for harmful outputs such as toxic or dishonest content (Crothers et al., 2023), among others. As it seems, the research on methods aimed at detecting the origin of a given text to mitigate the dissemination of forged content and to prevent technology-aided plagiarism still lag behind the rapid advancement of AI itself.[1]

Many works have focused on tools that could spot such AI-generated outputs and address these underlying risks. From a bird's eye view, this typically involves using *detector* models to discriminate *generator* models' outputs from legitimate human writings. Multiple versions of this generic text classification task have been considered, varying e.g. the number of possible categories to distinguish and the amount of supervision (see Section 5). Owing to its large user base and applications, the largest effort has focused on one specific generator, ChatGPT, for which training and test data is easily obtained. Yet, the corresponding supervised binary problem, with a unique known generator, is not the only way to frame this task. A more challenging problem, that we study here, is generator-agnostic artificial text detection, where the models to detect are not known in advance.

As pointed out e.g., in (Antoun et al., 2024; Hans et al., 2024; Wang et al., 2024a), the performance of artificial text detection systems varies depending on the choices of the detector(s) / generator(s) pair. The detector may serve to assess text probabilities, as in (Mitchell et al., 2023; Bao et al., 2024), or to regenerate texts, as e.g., in (Mao et al., 2024; Yang et al., 2024). This implies that the search for optimal detection performance should include a systematic exploration of the space of possible detectors. As the number and diversity of LLMs keep increasing, such exploration seems not only challenging but also unrealistic. Furthermore, Dugan et al. (2024) demonstrated that the current detection methods are brittle and easily fooled by simply changing the generator or the associated sampling method, which means that the optimal detector may need to be periodically revised.

---

[1]As illustrated by the discontinuation of OpenAI's detector `https://openai.com/index/new-ai-classifier-for-indicating-ai-written-text/`.

All this suggests that more efforts are needed to increase the robustness of existing detectors to changes in the generation method. For this, our proposal relies on *ensemble methods*, where a coalition of several models can be exploited to build the detector. For this, we generalize perplexity-based approaches, which flag as "artificial" texts that have a suspiciously small perplexity. As perplexity is also an encoding measure, our combination algorithm will seek to identify time-varying mixture models to minimize the worst-case expected encoding size. This also corresponds to the combination leading to the highest mutual information, that we implement with an architecture depicted in Figure 1. Further details, explanations, and proofs can be found in section 2. This approach eliminates the need to empirically search for the best detector(s), and yields detection systems that can robustly detect multiple generators. Furthermore, it lends itself to the smooth enrichment of the ensemble as new models become available, thereby improving generalization to unseen models.

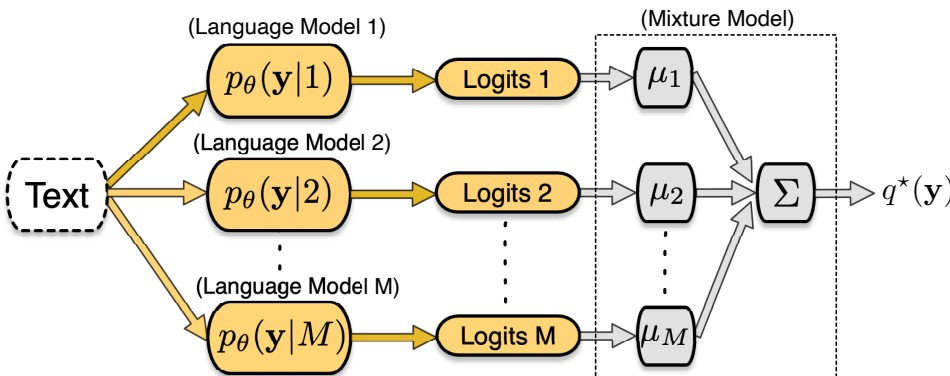

Figure 1: Mixture Model. $\{\mu_i\}$ are weights associated to models in the mixture, defined in Equation 4.

**Our contributions.** In this paper, using fundamental information-theoretic principles from universal compression, we derive a new ensemble score that optimally combines the strength of multiple LLMs to detect forged texts. Our experiments use both standard benchmarks comprising multiple domains and languages, as well as a new corpus of machine-generated texts. They confirm that ensembling strong LLMs yields detectors that can robustly identify a multiplicity of generators and that compare favorably with several recent proposals using a predefined set of detector models. Our analyses explore the effect of incrementally adding models into the ensemble, and also highlight, using a multilingual dataset, how the contribution of each constituent model changes when the source of artificial texts is modified.

## 2 THE MOSAIC APPROACH

### 2.1 BACKGROUND

We consider models for language generation tasks that define a probability distribution over strings. Formally, language models are probability distributions over an output space $\mathcal{Y}$ which contains all possible strings over vocabulary $\Omega$: $\mathcal{Y} \triangleq \{\text{BOS} \circ \mathbf{y} \circ \text{EOS} \mid \mathbf{y} \in \Omega^*\}$, BOS and EOS denote respectively the beginning-of-sequence and end-of-sequence tokens, and $\Omega^*$ is the Kleene closure of $\Omega$.

Today's models for language generation are typically parameterized by encoder-decoder or decoder-only architectures with attention mechanisms with trainable weights $\theta \in \Theta$. These models follow a local-normalization scheme, meaning that $\forall\, t > 0$, $p_\theta(\cdot|\mathbf{y}_{<t},)$ defines a conditional probability distribution over $\bar{\mathcal{Y}} = \mathcal{Y} \cup \text{EOS}$. The probability of a sequence $\mathbf{y} = \langle y_0, y_1, \ldots \rangle$ is expressed as:

$$p_\theta(\mathbf{y}) = \prod_{t=1}^{T} p_\theta(y_t|\mathbf{y}_{<t}), \tag{1}$$

and $\mathbf{y}_{<t} = \langle y_0, \ldots, y_{t-1} \rangle$, with $y_0 = \text{BOS}$.

**Measuring information.** The fundamental concept in information is "surprisal", using the relationship: information $= -\log(\text{probability})$ (Cover and Thomas, 2006), and assuming the use of

coding techniques such as Huffman and Arithmetic codes (Shields, 1996) which give message lengths closely approximating the ideal length in binary digits. Here, the measure of information is most conveniently introduced in the context of lossless compression. That is, we will look a what happens when information is passed from a *encoder* to a *decoder*. Humans often use codes, such as natural languages, which are not optimal for any set of prior expectations. There are good reasons for natural languages to be less than "optimal" when regarded as codes for data compression. One reason is that spoken language is transmitted from speaker to listener via a noisy channel. Codes with substantial redundancy can tolerate some degree of corruption without becoming unintelligible, whereas optimal compression codes, in which every digit matters, are very sensitive to corruption. Artificial codes for information storage are often designed so that legal strings conform to a strict pattern while most strings do not. A corrupted received string can then be corrected by replacing it with the nearest legal string, provided the degree of corruption is not too great. However, for our purposes, we need not consider errors in the storage of messages, nor be concerned with error-correcting codes.

**Explanations of data.** Given a body of text represented in a finite string $\mathbf{y}_{<t} = \langle y_0, \ldots, y_{t-1}\rangle$, an "explanation" of the next token $y_t$ is a binary string encoding the symbol in a particular format with *minimum* length $\mathcal{L}_\theta(y_t|\mathbf{y}_{<t}) \triangleq -\log p_\theta(y_t|\mathbf{y}_{<t})$. Its expected value is termed **conditional entropy**:

$$\mathcal{H}_\theta(Y_t|\mathbf{y}_{<t}) = \sum_{y_t \in \Omega} p_\theta(y_t|\mathbf{y}_{<t})\mathcal{L}_\theta(y_t|\mathbf{y}_{<t}).$$

Finally, another important concept is the **conditional mutual information** (MI) between two random variables $\mathbb{M}$ and $Y_t$, given a sequence value $\mathbf{y}_{<t}$, defined as (Cover and Thomas, 2006):

$$\mathcal{I}_\theta(\mathbb{M}; Y_t|\mathbf{y}_{<t}) = \mathcal{H}_\theta(Y_t|\mathbf{y}_{<t}) - \mathcal{H}_\theta(Y_t|\mathbb{M}, \mathbf{y}_{<t}),$$

$$\mathcal{H}_\theta(Y_t|\mathbb{M}, \mathbf{y}_{<t}) = \mathbb{E}_{m\sim\mu(m|\mathbf{y}_{<t})}\left[\mathcal{H}_\theta(Y_t|m, \mathbf{y}_{<t})\right].$$

It captures the amount of information we get about $\mathbb{M}$ when observing $Y_t$, and already knowing $\mathbf{y}_{<t}$.

## 2.2 COMBINING LLMS

Let $\mathcal{P}_\mathcal{M}(\mathcal{Y}) \triangleq \{p_\theta(\mathbf{y}|m) : m \in \mathcal{M}\}$ be a family of LLMs, as presented in (1), with identifying set of indexes $\mathcal{M} = \{1, \ldots, M\}$. Given $\mathbf{m}$ a sequence of $T$ indexes in $\mathcal{M}$, where $m_t$ specifies the model index for generating token $y_t$, we derive:

$$p_\theta(\mathbf{y}|\mathbf{m}) = \prod_{t=1}^{T} p_\theta(y_t|m_t, \mathbf{y}_{<t}). \tag{2}$$

Depending on the choice of explanation $m_t$ for token $t$, certain tokens in $\Omega$ become unsurprising (high probability) while others become very surprising or unbelievable (low or zero probability).

A family of LLMs can be exploited to produce explanations of token sequences. To this end, we assume some $\hat{m}_t = f_t(\mathbf{y}_{<t})$ which selects a probability distribution $p_\theta(y_t|\hat{m}_t, \mathbf{y}_{<t})$ over $\Omega$. Given $\hat{m}_t$, the encoder can construct an optimum code for token $y_t$, using distribution $p_\theta(y_t|\hat{m}_t, \mathbf{y}_{<t})$. Therefore, a rich family of LLMs allows us to capture and represent regular patterns in token sequences via the model selector $\hat{m}_t$ and subsequently use it to minimize the total expected codelength.

**Identifying explanations of data.** We now turn to the problem of determining an adequate sequence of models $\hat{\mathbf{m}} = \langle \hat{m}_0, \ldots, \hat{m}_T\rangle$. Our goal will be to derive a robust scoring algorithm that best extracts regularity in the data, which is equivalent to identifying **the model that achieves the best compression of the input tokens**. Suppose we are given a family of LLMs $\mathcal{P}_\mathcal{M}(\mathcal{Y})$, with corresponding Shannon codelengths $\mathcal{L}_\theta(y_t|m, \mathbf{y}_{<t}) \triangleq -\log p_\theta(y_t|m, \mathbf{y}_{<t})$ for each $y_t$. These can be viewed as a collection of data compressors, indexed by $m$. We can measure the performance of encoding $y_t$ at time $t$ relative to $\mathcal{P}_\mathcal{M}(\mathcal{Y})$. If we chose to encode the token $y_t$ with model $q(y_t|\mathbf{y}_{<t})$, the resulting expected excess codelength (or overhead) w.r.t. any distribution $p_\theta \in \mathcal{P}_\mathcal{M}(\mathcal{Y})$ is:

$$\mathcal{R}_\theta(m, q\,; \mathbf{y}_{<t}) \triangleq \mathbb{E}_{y_t\sim p_\theta(y_t|m,\mathbf{y}_{<t})}\left[-\log q(y_t|\mathbf{y}_{<t})\right] - \mathcal{H}_\theta(Y_t|m, \mathbf{y}_{<t})$$

which is non-negative since $\mathcal{H}_\theta(Y_t|m, \mathbf{y}_{<t})$ is the *minimum expected codelength*. $\mathcal{R}_\theta$ represents the extra averaged number of bits needed to encode $y_t$ using the code/LLM $q(y_t|\mathbf{y}_{<t})$, as compared to

$\mathcal{H}_\theta(Y_t|m, \mathbf{y}_{<t})$, the number of bits needed if we would have used the best fitting LLM in $\mathcal{P}_\mathcal{M}(\mathcal{Y})$ with hindsight. However, the encoder cannot know the underlying model artificially generating $y_t$ so we take a worst-case approach and look for universal LLMs with small worst-case expected overhead, where the worst-case is over all models in $\mathcal{P}_\mathcal{M}(\mathcal{Y})$. $\mathcal{R}_\theta$ is our quality measure and hence, the 'optimal' LLM relative to $\mathcal{P}_\mathcal{M}(\mathcal{Y})$, for a given context $\mathbf{y}_{<t}$, is the distribution minimizing:

$$q^\star(y_t|\mathbf{y}_{<t}) \triangleq \underset{q \in \mathcal{P}(\Omega)}{\operatorname{argmin}} \ \underset{m \in \mathcal{M}}{\max} \ \mathcal{R}_\theta(m, q \, ; \mathbf{y}_{<t}), \tag{3}$$

where the minimum is over all distributions on $\Omega$. The minimizer is the code with the smallest overhead (extra number of bits) compared to the optimal code that is best in hindsight in the worst-averaged case over all LLMs in $\mathcal{P}_\mathcal{M}(\mathcal{Y})$.

**Leveraging codelengths for identifying AI-generated text.** The averaged overhead of the optimal codelength $-\log q^\star(y_t|\mathbf{y}_{<t})$ obtained by solving Eq. (3) seems to be a very reasonable choice for building a robust score function to detect AI-generated text because of the following properties:

- The better the best-fitting LLM in $\mathcal{P}_\mathcal{M}(\mathcal{Y})$ fits the artificially generated data, the shorter the codelengh $\mathcal{L}^\star(y_t|\mathbf{y}_{<t}) \triangleq -\log q^\star(y_t|\mathbf{y}_{<t})$.

- No LLM in $\mathcal{P}_\mathcal{M}(\mathcal{Y})$ is given a prior preference over any other since $\mathcal{R}_\theta(m, q^\star \, ; \mathbf{y}_{<t}) \leq \mathcal{R}_\theta(m, p_\theta \, ; \mathbf{y}_{<t})$ for all $p_\theta \in \mathcal{P}_\mathcal{M}(\mathcal{Y})$, i.e., we are treating all LLMs within our universe $\mathcal{P}_\mathcal{M}(\mathcal{Y})$ on the same footing.

These observations lead to the following score.

**Definition 1** (MOSAIC Score). For an input sentence $\mathbf{w} = \langle w_0, w_1, \ldots \rangle$, the MOSAIC score is defined as:

$$S_{\text{Av}}(\mathbf{w}) \triangleq \frac{1}{TM} \sum_{t=1}^{T} \sum_{m \in \mathcal{M}} \Big[ \underbrace{\sum_{y_t \in \Omega} \mathbb{1}[y_t = w_t]\mathcal{L}^\star(y_t|\mathbf{w}_{<t})}_{\text{(codelength for observed token)}} - \underbrace{\sum_{y_t \in \Omega} p_\theta(y_t|m, \mathbf{w}_{<t})\mathcal{L}^\star(y_t|\mathbf{w}_{<t})}_{\text{(codelength for generated tokens from model 'm')}} \Big],$$

where $\mathcal{L}^\star(y_t|\mathbf{w}_{<t}) \triangleq -\log q^\star(y_t|\mathbf{w}_{<t})$. For a suitable $\delta > 0$, if $S_{\text{Av}}(\mathbf{w}) \geq \delta$, then the text $\mathbf{w}$ is declared to be human and otherwise AI-generated.

*Remark* 1. The first term in $S_{\text{Av}}(\mathbf{w})$ represents the averaged per-token codelength of the input sequence for the code/LLM $q^\star$, which corresponds to the well-known perplexity. The second term is the averaged per-token codelength over all randomly generated sequences according to the averaged LLMs in $\mathcal{P}_\mathcal{M}(\mathcal{Y})$, which is the average of the cross-entropy with respect to all models in the family. The resulting score is the difference between these codelengths. If the input sentence is generated by one of the LLMs in the family or another closely related one, the score is expected to be small, as $q^\star$'s goal is to extract as much regularity as possible from $\mathbf{w}$. However, if the input sentence is human-generated, the score is expected to be large as the first term will dominate.

The next proposition provides a theoretical result together with an efficient iterative algorithm to optimally solve expression (3). The proof of this proposition is relegated to Appendix A.

**Proposition 1** (Optimal codelength). *The optimal solution to* (3) *is a mixture of LLMs:*

$$q^\star(y_t|\mathbf{y}_{<t}) = \sum_{m \in \mathcal{M}} \mu^\star(m|\mathbf{y}_{<t})p_\theta(y_t|m, \mathbf{y}_{<t}),$$

*where the distribution* $\mu^\star(\cdot|\mathbf{y}_{<t})$ *of the random variable* $\mathbb{M}$ *over LLM indice in* $\mathcal{M}$ *satisfies:*

$$\mu^\star(\cdot|\mathbf{y}_{<t}) \triangleq \underset{\mu \in \mathcal{P}(\Omega)}{\operatorname{argmax}} \ \mathcal{I}_\theta\big(\mathbb{M}; Y_t|\mathbf{y}_{<t}\big). \tag{4}$$

*Furthermore, the weights* $\{\mu^\star(m|\mathbf{y}_{<t})\}_{m \in \mathcal{M}}$ *can be computed efficiently with the Blahut–Arimoto algorithm, and are referred to as Blahut–Arimoto weights.*

## 2.3 IMPLEMENTATION

Proposition 1 implies that to implement the scoring function introduced in Definition 1, it would be enough to solve the optimization in Eq. (4), which is much simpler than Eq. (3). Interestingly, Blahut–Arimoto algorithm (Arimoto, 1972; Blahut, 1972) provides us with an efficient iterative method to compute the maximization of mutual information in Eq. (4) (see Appendix B). This algorithm lies at the core of our scoring procedure.

---

**Algorithm 1** MOSAIC Scoring

1: **Input**: text $\mathbf{w} = \langle w_0, w_1, \ldots \rangle$, LLMs $(1, 2, \ldots, M)$
2: **for** $w_t$ in $\mathbf{w}$ **do**
3: $\quad \mu^*(m|\mathbf{w}_{<t}) \leftarrow$ Blahut–Arimoto $(\mathcal{P}_\mathcal{M}(\mathcal{Y}); \mathbf{w}_{<t})$
4: $\quad q^*(y_t|\mathbf{w}_{<t}) \leftarrow \sum_{m \in \mathcal{M}} \mu^*(m|\mathbf{w}_{<t}) p_\theta(y_t|m, \mathbf{w}_{<t})$
5: $\quad s_t(\mathbf{w}) \leftarrow \mathcal{L}^\star(w_t|\mathbf{w}_{<t}) - \frac{1}{M} \sum_{m \in \mathcal{M}} \left( \mathbb{E}_{y_t \sim p_\theta(y_t|m, \mathbf{w}_{<t})} \left[ \mathcal{L}^\star(y_t|\mathbf{w}_{<t}) \right] \right)$
6: **end for**
7: $S_{\text{Av}}(\mathbf{w}) \leftarrow \frac{1}{T} \sum_t s_t(\mathbf{w})$                      ▷ MOSAIC score for the whole text

---

# 3 EXPERIMENTAL SETTINGS

## 3.1 DATASETS & METRICS

We evaluate our method on a diverse set of texts and generative models from the literature: RAID (Dugan et al., 2024), Ghostbuster (Verma et al., 2023), Binoculars (Hans et al., 2024), M4 (Wang et al., 2024a) and a corpus of scholarly texts (Liyanage et al., 2022).

**RAID** contains about 15k natural texts in English from a variety of domains; the artificial part version contains approximately 500K, generated with a diverse set of recent models, also varying the sampling procedure. As the test set is not publicly released, we select a balanced random subset of 2000 texts for our experiments. RAID also includes an artificially noised subcorpus, which was not used in our experiments.

**The Ghostbuster dataset** is split into three parts: WritingPrompts, based on the r/WritingPrompts subreddit where users submit stories in response to short prompts; Reuters, using the Reuters 50-50 authorship identification dataset (Houvardas and Stamatatos, 2006); and Essays, comprising essays scraped from IvyPanda,[2] a website dedicated to homework help. Each part contains 1,000 original texts that have been regenerated from their headlines[3] with ChatGPT (using 5 different prompts) and with Claude (Anthropic, 2023) (one prompt), for a total of 7,000 texts.

**The Binoculars dataset** contains samples of human-written texts from CCNews, Pubmed and CNN; alternative completions are automatically generated using a Llama-2-13b (Touvron et al., 2023) and Falcon-7b (Almazrouei et al., 2023). Their generation technique uses the first 50 tokens of each text as a prompt to generate a machine output. Those first 50 tokens are then removed from the result so that samples only contain machine-generated texts.[4]

**The M4[5] corpus** is a massive dataset of natural texts collected from a diverse set of sources. Comparable artificial texts are generated by 6 LLMs, with prompts such as article titles, headlines, or abstracts depending on their domain. In our experiments, we only use one "multilingual" generator (ChatGPT, https://chatgpt.com/), and the balanced sets made of $3,000$ pairs of (artificial, natural) texts in Russian (Rus), Bulgarian (Bul), Arabic (Ara), and German (Ger).

**The academic benchmark** proposed by Liyanage et al. (2022) is generated using a GPT2 model (Radford et al., 2019) fine-tuned on papers from Arxiv's "Computation and Language" (CS.CL) domain. 100 seed texts are used to generate new papers of comparable length, using the first 50 words

---

[2]https://ivypanda.com/
[3]For Reuters and Essays, the headlines were themselves generated based on the text.
[4]For most texts, only one single artificial text (using either Llama or Falcon) is available.
[5]For *Multi-Lingual, Multi-Domain, Multi-Generator Machine-Generated text*.

Table 1: Natural and Artificial texts used in the experiments, all lengths are in Llama-2 tokens.

| Corpus Name | Generator(s) | Human | | Artificial | |
|---|---|---|---|---|---|
| | | # texts | avg length | # texts | avg length |
| RAID | Multiple LLMs | 2,000 | 452 | 2,000 | 353 |
| Binoculars | Llama-2-13b, Falcon | 9,148 | 2,252 | 11,178 | 677 |
| Binoculars+ | Llama-2-7b, Mistral-7b | 9,148 | 2,202 | 18,295 | 397 |
| Ghostbuster | ChatGPT ($\times 5$), Claude | 1,000 | 826 | 6,000 | 754 |
| arxiv-cs.cl | GPT2 | 100 | 1,940 | 100 | 1977 |
| M4 (Multilingual) | ChatGPT | 12,000 | 729 | 12,000 | 649 |

as prompts. Sections such as "methodology", "results", "evaluation", and "discussion" are voluntarily omitted from texts, to ensure that discriminating factors do not rely on the comparative use of diagrams, tables, and equations.

Table 1 displays the main statistics for these 5 corpora. For completeness, we also augment the Binoculars dataset using Llama-2-7b and Mistral-7b as alternative text generators with Huggingface's transformers `model.generate()` (Wolf et al., 2020).[6] As for the original corpus, the first 50 tokens of the original texts provide the starting context for both machine & human texts. To also test our method on extreme cases, we randomly generated 3,000 texts of 500-token using a unigram model trained on the Brown Corpus (Francis and Kucera, 1979),[7] along with original extracts of the same length. These corpora will be released with our detector's implementation.

These datasets represent a large variety of genres, themes, languages, sampling strategies, and generators, allowing us to thoroughly assess the robustness of our detection strategy. Using Binoculars+, we can evaluate detection performance for texts produced by one of our detectors.

**Metrics.** As in most studies, we report the AUROC score as our main evaluation metric. Depending on the application, True Positive Rate (TPR) for a predefined False Positive rate (e.g., 5%) is also worth looking at and is also reported in most of our results. All these scores are obtained using scikit-learn (Pedregosa et al., 2011). It is important to note that our experiments involve imbalanced settings in the case of Ghostbuster and the original Binoculars datasets, see numbers in Table 1, whereas, for our Binoculars+ regenerations and the M4 multilingual datasets, the test data contains a balanced number of (original, generated) pairs of texts.

### 3.2 BASELINES

Machine-generated text detection methods are usually divided into two main categories (Dugan et al., 2024): *supervised* and *unsupervised* (metric-based). The former uses supervision data to fine-tune a pre-trained model for the detection task, typically focusing on some known generators. Our method belongs to the latter family, as the MOSAIC score of Eq. (2), which serves to discriminate forged content, does not require any training data. Accordingly, we compare our method to other zero-shot unsupervised techniques used for machine-generated text detection:

**Perplexity (PPL)** based detectors use a threshold on the text's log-perplexity, assuming that LLMs usually generate texts that have a lower perplexity than human's productions (see Vasilatos et al., 2023; Guo et al., 2023; Mitrović et al., 2023; Li et al., 2024 *inter alia*). This yields a very straightforward criterion for detecting machine-generated texts. We compute this baseline separately for all available models in our ensemble. Additionally, PPL (average) reports the performance obtained with an average of all perplexities within our ensemble.

**DetectGPT** (Mitchell et al., 2023) generates minor perturbations of a given text passage, then computes the differences in log-probability between the original text and its perturbed versions. DetectGPT relies on the observation that when slightly perturbed, the log-probability of artificial texts consistently decreases, which is not the case for human-generated texts. We used the default

---

[6]`https://huggingface.co/docs/transformers`. For Llama: nucleus sampling parameters: repetition_penalty: 1.18, temp.: 0.6, top_p: 0.9; for Mistral: repetition_penalty: 1.18, temp.: 0.7, top_p: 1.

[7]We tokenize the Brown Corpus (Francis and Kucera, 1979) with Llama tools and use count ratios as probability estimates, with Laplace smoothing ($\epsilon = 1e - 10$).

parameters in the authors' implementation, running the "10d" experiment with GPT2-medium computing the log-probabilities and T5-large (Raffel et al., 2020) generating the perturbations. Since the original implementation is not optimized for long sequences and imbalanced datasets, we split the lengthy texts into smaller chunks and downsampled the larger dataset to ensure both were of equal size. **FastDetectGPT** (Bao et al., 2024) relies on the same principles but uses a different approach to (quickly) sample the perturbations of the text. Its authors found that it was both faster and slightly better than the original implementation.

**Binoculars score** (Hans et al., 2024) is also based on log-probabilities: it compares the average log-probability of an input text for a detector model with the cross-entropy of an auxiliary model (see Eq. (23) in Appendix C). With this score, artificial texts should have a score lower than natural texts. We used the default model pair selected by the authors, chosen for their best results on the Binoculars dataset, with Falcon-7b-instruct as the detector model, used to compute the perplexity, and Falcon-7b as the auxiliary model. We also report detection results using Eq. (23) with two models from our ensemble.

Baseline scores in Table 2 are obtained with the implementations provided with the original papers. These results do not directly compare with those of Table 3 as the underlying set of models is different.

Table 2: Detection performance of baseline systems. AUROC scores.

| | RAID | Binoculars | | | Ghostbuster | | | M4 (multilingual) | | | | Scho. | Avg. |
|---|---|---|---|---|---|---|---|---|---|---|---|---|---|
| | | Pubmed | CNN | CCnews | Reuter | Essay | Reddit | Ara | Bul | Ger | Rus | | |
| DetectGPT | 0.632 | 0.666 | 0.635 | 0.571 | 0.714 | 0.916 | 0.757 | 0.576 | 0.589 | 0.524 | 0.597 | 0.440 | 0.635 |
| FastDetectGPT | 0.706 | 0.787 | 0.925 | 0.772 | 0.829 | 0.949 | 0.938 | 0.874 | 0.683 | 0.827 | 0.596 | 0.549 | 0.786 |
| Binoculars | 0.853 | 0.988 | 0.995 | 0.979 | 0.993 | 0.996 | 0.990 | 0.686 | 0.742 | 0.914 | 0.674 | 0.505 | 0.860 |

Table 3: Artificial text detection performance of detectors built with a fixed set of 4 models. Detection may involve running 1, 2, or 4 models. AUROC scores.

| | RAID | Binoculars | | | Ghostbuster | | | M4 (multilingual) | | | | Scho. | Avg |
|---|---|---|---|---|---|---|---|---|---|---|---|---|---|
| | | Pubmed | CNN | CCnews | Reddit | Reuter | Essay | Ara | Bul | Ger | Rus | | |
| Best single-model | 0.834 | **0.999** | **0.995** | **0.975** | 0.878 | 0.886 | 0.818 | **0.985** | **0.988** | 0.832 | **0.816** | **0.517** | 0.877 |
| Best two-model | 0.803 | 0.989 | 0.994 | 0.973 | 0.677 | 0.663 | 0.481 | 0.897 | 0.959 | 0.860 | 0.492 | 0.381 | 0.764 |
| avg PPL | 0.730 | 0.808 | 0.566 | 0.584 | 0.980 | 0.980 | 0.992 | 0.897 | 0.887 | 0.638 | 0.598 | 0.516 | 0.765 |
| $q^\star$ (log-probs) | 0.746 | 0.807 | 0.566 | 0.579 | **0.985** | **0.984** | **0.994** | 0.893 | 0.901 | 0.640 | 0.592 | 0.509 | 0.766 |
| MOSAIC-4 (avg) | **0.850** | 0.992 | 0.993 | 0.971 | 0.946 | 0.971 | 0.911 | 0.909 | 0.974 | 0.890 | 0.737 | 0.421 | **0.880** |
| MOSAIC-4 (unif) | 0.844 | 0.992 | **0.995** | **0.975** | 0.920 | 0.951 | 0.876 | 0.909 | 0.974 | **0.893** | 0.745 | 0.416 | 0.874 |

## 4 EXPERIMENTAL RESULTS

The following experimental results all use the same configuration. Unless explicitly stated otherwise, MOSAIC-4 uses an ensemble of models composed of TowerBase-7b, TowerBase-13b (Alves et al., 2024), Llama-2-7b-chat, and Llama-2-7b (Touvron et al., 2023). This choice of models is motivated by their shared tokenizer.

### 4.1 THE ROBUSTNESS OF ENSEMBLE METHODS

In our experiments, we first evaluate the added robustness of the various ensemble methods compared to using just one model. Given our ensemble detectors, we consider the following options: (a) PPL detection for each model; (b) (Fast)DetectGPT for each model; (c) Binoculars score for each pair of models; (d) PPL detection with average PPL scores or with the log-probabilities of the optimal distribution $q^*$;[8] (e) MOSAIC; (f) the MOSAIC score, using a uniform weighting scheme instead of Blahut-Arimoto weights. Methods (a,b) require just one model; (c) requires two; (d,e,f) require four. For the sake of space, we only report in Table 3 for methods (a) and (b) the model with the best performance on the CC_News subset of Binoculars; likewise, for (c) we select the best pair of models

---

[8]Strictly speaking, not a perplexity, as the Blahut-Arimoto weights change for each token.

on the same test corpus (TowerBase-7b as the detector, and Llama-2-7b as the auxiliary model). The complete set of results is in Table 5 in Appendix E.

We see a large variation across datasets and generation techniques: for some, a near-perfect detection can be achieved (notably Binoculars datasets, and to a lesser extent, Ghostbuster, for which PPL-based detectors can be remarkably good). However, varying the domain (Scholarly texts) and/or languages (M4) can have a detrimental effect on detection performance.

Among the baselines (Table 2), Binoculars is the most robust and achieves the best average performance, perhaps owing to the better underlying detector model (Falcon vs. GPT2). Binoculars is very sensible to changes in scripts, as the results on the M4 dataset for Arabic (Arabic script), Bulgarian and Russian (both written in Cyrillic), show.

Our best single-model detector (FastDetectGPT with Tower-13b, selected out of 12 alternatives) achieves very good average scores and outperforms the original Binoculars baseline. By contrast, our best implementation of the Binoculars score with 2 models from our ensemble (also selected out of 12 combinations), is less effective and lags behind the Falcon-based Binoculars detector. Using 4 models, MOSAIC(avg) yields the best on average, and dispenses with a prior search for the optimal configuration. For RAID, arguably the more challenging dataset, it compares with the best Binoculars model from Hans et al. (2024).

An interesting follow-up question is about the respective strength of each detector model: can this be measured using the Blahut-Arimoto weights? For our ensemble, we find that all models get a reasonable (and varying) share of these weights. We see however that Llama-7b instruct consistently receives very large weights (see Figure 4a in Appendix). For the multilingual tests from M4, we observe that the weights of the TowerBase models tend to increase, as compared to when just looking at English texts (see Figure 2 in Appendix). This illustrates the benefits of using complementary models, each one with its own domain of "expertise".

Scholarly texts pose another problem. As these were produced by a GPT-2 model fine-tuned on Arxiv's "computation and language" section, the distribution of tokens greatly differs from all models considered in this study. The generated texts therefore appear completely out-of-domain and get a high perplexity for all models, which makes them difficult to discriminate from human texts (see Figure 3 in the Appendix) with perplexity-based models. This holds similarly for our ensemble (e.g., MOSAIC-4) and for baseline models (e.g., Falcon-based Binoculars), which achieve respective AUROCs of 0.421 and 0.505. This shows that brittleness issues are not fully solved, and hints at including more domain-adapted models in detector ensembles.

Adversarial attacks, reported in Table 7, deteriorate slightly the performances of our method with the exception of "synonym replacement", which makes generated texts more surprising while not influencing human-produced texts in the same way (the idea behind DetectGPT), completely breaking our method. Interestingly, replacing some characters with homoglyphs actually improves our results, we hypothesize it is because the Tower models have seen Cyrillic in their training data.

### 4.2 INCLUDING THE GENERATOR IN THE DETECTOR ENSEMBLE

Table 4 reports results with Binoculars+, where we augment Binoculars with comparable texts generated with two models: Llama-7b, which is part of our ensemble, and Mistral-7b, which is not. For Pubmed and CC_news, we see the same trend: both the original artificial texts and the Mistral-generated texts are much easier to detect than Llama-generated texts. Contrarily, for CNN, Llama-7b is almost perfectly detected. Overall, having the generator inside the ensemble of detectors does not seem particularly advantageous.

However, it should be noted that the sampling method plays an important role in the performance of unsupervised detection methods, as displayed in Table 6 in the Appendix. Recall that our regeneration uses a temperature of 0.6 along with nucleus sampling ($p = 0.9$) and a repetition penalty of 1.18 (see Section 3.1), as recommended by the official Llama repository[9] as they produce more diverse, and in a way, more human-like texts than when using greedy decoding. This means that the actual sampling distribution does not fully match the base distribution of the detector, and consistently generates "surprising" tokens for the generator model.

---

[9]https://github.com/meta-llama/llama/blob/main/example_text_completion.py

Table 4: Identification results for the Binoculars and Binoculars+ datasets. Single models detectors are PPL-based. AUROC scores. Best scores in each column are in boldface.

| | Pubmed | | | CNN | | | CC news | | | Avg. |
|---|---|---|---|---|---|---|---|---|---|---|
| | Orig. | Llama | Mist. | Orig. | Llama | Mist. | Orig. | Llama | Mist. | |
| TowerBase-7b | 0.798 | 0.631 | 0.806 | 0.557 | 0.999 | 0.639 | 0.582 | 0.573 | 0.669 | 0.695 |
| TowerBase-13b | 0.823 | 0.528 | 0.801 | 0.561 | 0.999 | 0.641 | 0.578 | 0.439 | 0.652 | 0.669 |
| Llama-2-7b | 0.786 | 0.676 | 0.820 | 0.544 | **1.000** | 0.704 | 0.556 | 0.617 | 0.693 | 0.711 |
| Llama-2-7b-chat | 0.817 | 0.690 | 0.843 | 0.598 | **1.000** | 0.716 | 0.617 | 0.632 | 0.718 | 0.737 |
| $q^*$ (log-probs) | 0.807 | 0.641 | 0.836 | 0.566 | **1.000** | 0.727 | 0.579 | 0.568 | 0.703 | 0.714 |
| MOSAIC-4 (avg) | **0.992** | **0.887** | **0.961** | **0.993** | 0.999 | **0.971** | **0.971** | **0.854** | **0.940** | **0.952** |

In a follow-up experiment, we replace the distribution computed by Llama-2-7b with a "distorted" version, which approximates the effects of temperature and nucleus-sampling[10]. This has a clear effect on the Blahut–Arimoto weights defining $q^*$, which increase for Llama-2-7b and decrease for Llama-2-7b-instruct, but hardly change the detection performance (see Figure 4 in Appendix).

### 4.3 AUGMENTING THE ENSEMBLE

**With a Strong Model.** MOSAIC makes the augmentation of the ensemble quite easy, as long as all models use the same underlying tokenizer. To showcase the effect of this feature, we add Phi-3 (Abdin et al., 2024) in our ensemble and experiment with Binoculars and Ghostbuster datasets. In this experimental setting, this extension of the ensemble is of little consequence for the former test, and provides clear gains for the latter (see lines "+phi" in Appendix, Table 9). Accordingly, we observe that Phi consistently gets a substantial share of the Blahut–Arimoto weights (see Figure 5 in Appendix) and plays a significant role in the classification decision.

**With a Weak Model.** As noted by Hans et al. (2024), simple random generators are hard to detect for Binoculars (as well as for detectors using a PPL-based threshold): this is because random word salads are "surprising" (have a high perplexity) for well-trained detectors, and tend to be confused with human productions. We reproduce this observation using the corpus generated with a unigram model and observe that all baselines, as well as our proposed detector, achieve AUROC scores close to 0 for this dataset. The PPL of such "word salads" is however much larger than for human texts, so setting an upper bound of the human PPL would, in that case, provide a very easy fix.

It is tempting to see whether adding such a poor generator into our combination would be of any help. For this experiment, we reuse the generator unigram language model and combine it with our four baseline detectors. The detector obtained with this extended ensemble remains unable to sort random from human texts: having the random model in the MOSAIC algorithm does not make random texts more likely. The added unigram model is also slightly detrimental for detecting strong generators, as we observe a mild drop in performance compared to using just 4 models (see Appendix, Table 9). This is because the unigram model predicts unexpected tokens most of the time and therefore often gets a significant weight in the Blahut–Arimoto weights (see Figure 6 in Appendix), which leads to a huge discrepancy between the cross-entropies scores that are averaged in Definition 1. However, this effect remains small; even with this weak model added, our ensemble detector remains rather strong.

## 5 RELATED WORK

The improved text generation abilities of LLMs raise concerns about potential misuses such as disinformation (Zellers et al., 2019), abusive content (Crothers et al., 2023), forged academic publications (Liu et al., 2024), or cheating during exams (Vasilatos et al., 2023). Since such fake texts seem difficult for humans to spot (Gehrmann et al., 2019), the issue of automatically detecting machine-generated texts has been subject to an increasing focus. This problem can be framed as a binary human vs. non-human decision, as the problem of detecting one known artificial agent (e.g., ChatGPT (Mitrović et al., 2023; Liu et al., 2024)), or as discriminating the correct model in a predefined list (Li

---

[10]We divide the model logits by the temperature, then apply softmax and perform nucleus filtering with $p = 0.9$. The resulting distribution is smoothed so out-of-nucleus tokens still get a small positive value.

et al., 2023). Another distinction is between closed-domain (e.g. scientific (Liyanage et al., 2022), academic (Liu et al., 2024) or user-generated content (Fagni et al., 2021; Kumarage et al., 2023)) vs. open-domain text detection. Assuming the generator models are known, various settings can be considered, depending on whether models can be openly queried (open parameters), whether they expose their full logits, or just the top prediction (and associated probability), etc.

Supervised detection with a single generator often achieves detection rates in the high 90s (Zellers et al., 2019; Guo et al., 2023; Liu et al., 2024), using classifiers based on Roberta (Conneau et al., 2020) or T5 (Raffel et al., 2020). However, these approaches are brittle and their success depends on particular generator-detector pairs (Antoun et al., 2024), prompting e.g. Verma et al. (2023) to design automatic feature extractors from multiple detectors to improve the robustness of their system.

Unsupervised detection is more challenging. Most approaches rest on the idea that human-written texts are more "surprising" than artificial texts[11], leading to a difference in token-wise log-probability[12]. This idea is used in GPTzero[13] and thresholding perplexity usually provides strong baselines (see, *inter alia*, (Gehrmann et al., 2019; Ippolito et al., 2020; Mitchell et al., 2023)). Such techniques heavily rely on the *detector* model(s) used to compute the log-probabilities of input texts, which must be robust to variations in domains, genres, styles, and languages (Wang et al., 2024a); and to variations in the generator itself (Antoun et al., 2024).

Mitchell et al. (2023) and Bao et al. (2024) exploit a similar intuition, arguing that small random perturbations of an artificial text will on average make it less likely, unlike human-written texts. They derive a statistical criterion based on the curvature of the log-probability function, and report near-perfect detection scores on three types of texts, generated by 5 models. The Binoculars score of Hans et al. (2024) also relies on a function of the per-token log-perplexity, contrasted with the cross-entropy of an auxiliary model.

These valuable works point to **the over-reliance on one specific detector model as a major limitation of the state-of-the-art.** Our proposed mitigation relies on ensemble techniques, that are also considered in the supervised detection setting, e.g. in (Verma et al., 2023; Wang et al., 2023; El-Sayed and Nasr, 2023; Liyanage and Buscaldi, 2023).

Abandoning generator-detector-based techniques altogether, (Mao et al., 2024; Yang et al., 2024) develop effective detection approaches based on regeneration, prompting the (known) generator to regenerate part of the input text. The intuition is that artificial inputs are likely to be regenerated exactly, while **human texts exhibit greater redundancy, resulting in the need for longer code-lengths.** Other strategies include text watermarking (Kirchenbauer et al., 2023a;b; Liu and Bu, 2024), though its efficiency and robustness are still subject to discussions, e.g., (Zhang et al., 2023).

Recent works focus on detection robustness. Wang et al. (2024b) find that after simple modifications, only watermarking remains able to accurately identify artificial documents. Dugan et al. (2024) present artificial texts generated with multiple models and sampling strategies, additionally subject to various adversarial attacks, observing that most detectors suffer large drops in performance. In their comparison, Binoculars (Hans et al., 2024) stands out, achieving decent detection scores at False Positive Rates under 1%.

## 6 SUMMARY AND DISCUSSION

Our MOSAIC method effectively harnesses the ensemble's strength, achieving great results across datasets and languages, eliminating the need to find the best detector while offering a scalable solution that can incorporate future models. However, it is currently computationally costly, as each model must run on the text (while not the goal of this work, some improvements are proposed in Section D). We also need to develop information-theoretic tools to select the most useful models and filter out less effective ones, as shown with the unigram model. Similarly, the proximity of models needs further study, as we suspect our detectors are too far from the fine-tuned model's generating distribution used to create the academic dataset, hence why it evades detection.

---

[11]Assuming generation does not use random sampling, in which case the reverse is likely to be observed, as long artificial texts drift away from natural writings (Zellers et al., 2019).

[12](Mitchell et al., 2023) argues that the difference is better seen at the level of log-ranks.

[13]https://gptzero.me/

## 7 ETHICS STATEMENT

It should be acknowledged that these tools are not infallible and consequently should not be used as the sole basis for punitive actions or decisions that could affect individuals or organizations. Such methods must be complemented by human oversight and verification before taking any drastic measure to ensure fairness. Moreover, in the course of this work, we have generated medical information and news articles using datasets "Pubmed", "CNN" and "CC_News" to test our method. While we intend to distribute these texts along with our implementation, we must enforce that those are only intended for research use and by no means should be circulated outside of this context, nor be presented as factual content.

## 8 REPRODUCIBILITY STATEMENT

All our experiments use public benchmarks and open-source code. The additional corpora created for this study, as well as all the code used to implement the MOSAIC algorithm, will also be openly released.

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

## A  PROOF OF PROPOSITION 1

*Proof.* We need to show the fundamental identity:

$$\Gamma(\mathbf{y}_{<t}) \triangleq \min_{q \in \mathcal{P}(\Omega)} \max_{m \in \mathcal{M}} \mathcal{R}_\theta(m, q\,;\mathbf{y}_{<t}) \tag{5}$$

$$= \max_{\mu \in \mathcal{P}(\mathcal{M})} \mathcal{I}(\mathbb{M}; Y_t | \mathbf{y}_{<t}), \tag{6}$$

where the optimal $q^\star(y_t|\mathbf{y}_{<t})$ achieving the minimum is characterized by the mixture:

$$q^\star(y_t|\mathbf{y}_{<t}) = \sum_{m \in \mathcal{M}} \mu^\star(m|\mathbf{y}_{<t}) p_\theta(y_t|m, \mathbf{y}_{<t}), \tag{7}$$

and the distribution $\mu^\star(m|\mathbf{y}_{<t})$ of the random variable $\mathbb{M}$ on $\mathcal{M}$ follows by solving:

$$\mu^\star(m|\mathbf{y}_{<t}) \triangleq \operatorname*{argmax}_{\mu \in \mathcal{P}(\Omega)} \mathcal{I}\big(\mathbb{M}; Y_t | \mathbf{y}_{<t}\big). \tag{8}$$

To this end, we start from the definition $\mathcal{R}_\theta$:

$$\mathcal{R}_\theta(m, q\,;\mathbf{y}_{<t}) \triangleq \mathop{\mathbb{E}}_{y_t \sim p_\theta(y_t|m,\mathbf{y}_{<t})} [-\log q(y_t|\mathbf{y}_{<t})]$$

$$- \min_{p_\theta \in \mathcal{P}_\mathcal{M}(\mathcal{Y})} \mathop{\mathbb{E}}_{y_t \sim p_\theta(y_t|m,\mathbf{y}_{<t})} \Big[ -\log p(y_t|\mathbf{y}_{<t}) \Big] \tag{9}$$

$$= \mathop{\mathbb{E}}_{y_t \sim p_\theta(y_t|m,\mathbf{y}_{<t})} [-\log q(y_t|\mathbf{y}_{<t})] - \mathcal{H}_\theta(Y_t|m, \mathbf{y}_{<t}) \tag{10}$$

$$= \mathcal{D}_{\mathrm{KL}}\Big( p_\theta(Y_t|m, \mathbf{y}_{<t}) \big\| q(Y_t|\mathbf{y}_{<t}) \Big), \tag{11}$$

where $\mathcal{D}_{\mathrm{KL}}(\cdot\|\cdot)$ denotes the Kullback–Leibler divergence. Hence, we can formally state our problem as follows:

$$\Gamma(\mathbf{y}_{<t}) = \min_{q \in \mathcal{P}(\Omega)} \max_{m \in \mathcal{M}} \mathcal{R}_\theta(m, q\,;\mathbf{y}_{<t})$$

$$= \min_{q \in \mathcal{P}(\Omega)} \max_{m \in \mathcal{M}} \mathcal{D}_{\mathrm{KL}}\Big( p_\theta(\cdot|m, \mathbf{y}_{<t}) \big\| q(\cdot|\mathbf{y}_{<t}) \Big)$$

$$= \min_{q \in \mathcal{P}(\Omega)} \max_{\mu \in \mathcal{P}(\mathcal{M})} \mathop{\mathbb{E}}_{m \sim \mu} \mathcal{D}_{\mathrm{KL}}\Big( p_\theta(\cdot|m, \mathbf{y}_{<t}) \big\| q(\cdot|\mathbf{y}_{<t}) \Big), \tag{12}$$

where the minimum is taken over all the possible distributions $q \in \mathcal{P}(\Omega)$, representing the expected value of regret of $q$ w.r.t. the worst-case distribution over $\mu \in \mathcal{P}(\mathcal{M})$. Notice that this is equivalent to the *average worst-case regret* Barron et al. (1998); Silva and Piantanida (2022). The equality in (12) holds by noticing that

$$\max_{\mu \in \mathcal{P}(\mathcal{M})} \mathop{\mathbb{E}}_{m \sim \mu} \mathcal{D}_{\mathrm{KL}}\Big( p_\theta(\cdot|m, \mathbf{y}_{<t}) \big\| q(\cdot|\mathbf{y}_{<t}) \Big) \le \max_{m \in \mathcal{M}} \mathcal{D}_{\mathrm{KL}}\Big( p_\theta(\cdot|m, \mathbf{y}_{<t}) \big\| q(\cdot|\mathbf{y}_{<t}) \Big) \tag{13}$$

and moreover,

$$\max_{m \in \mathcal{M}} \mathcal{D}_{\mathrm{KL}}\Big( p_\theta(\cdot|m, \mathbf{y}_{<t}) \big\| q(\cdot|\mathbf{y}_{<t}) \Big) = \mathop{\mathbb{E}}_{m \sim \widetilde{\mu}} \mathcal{D}_{\mathrm{KL}}\Big( p_\theta(\cdot|m, \mathbf{y}_{<t}) \big\| q(\cdot|\mathbf{y}_{<t}) \Big) \tag{14}$$

by choosing the measure $\widetilde{\mu}$ to be an uniform probability over the set $\widetilde{\mathcal{M}}$, which is defined as the set of maximizers:

$$\widetilde{\mathcal{M}} = \operatorname*{argmax}_{m \in \mathcal{M}} \mathcal{D}_{\mathrm{KL}}\Big( p_\theta(\cdot|m, \mathbf{y}_{<t}) \big\| q(\cdot|\mathbf{y}_{<t}) \Big),$$

and zero otherwise.

The convexity of the KL-divergence allows us to rewrite expression (12) as follows:

$$\min_{q \in \mathcal{P}(\Omega)} \max_{\mu \in \mathcal{P}(\mathcal{M})} \mathop{\mathbb{E}}_{m \sim \mu} \mathcal{D}_{\mathrm{KL}}\Big( p_\theta(\cdot|m, \mathbf{y}_{<t}) \big\| q(\cdot|\mathbf{y}_{<t}) \Big) = \max_{\mu \in \mathcal{P}(\mathcal{M})} \min_{q \in \mathcal{P}(\Omega)} \mathop{\mathbb{E}}_{m \sim \mu} \mathcal{D}_{\mathrm{KL}}\Big( p_\theta(\cdot|m, \mathbf{y}_{<t}) \big\| q(\cdot|\mathbf{y}_{<t}) \Big).$$

$$\tag{15}$$

This follows by considering a zero-sum game with a concave-convex mapping defined on a product of convex sets. The sets of all probability distributions $\mathcal{P}(\mathcal{M})$ and $\mathcal{P}(\Omega)$ are two nonempty convex sets, bounded and finite-dimensional. On the other hand, $(\mu, q) \to \mathbb{E}_{m \sim \mu} \mathcal{D}_{\mathrm{KL}}\Big(p_\theta(\cdot|m, \mathbf{y}_{<t})\big\|q(\cdot|\mathbf{y}_{<t})\Big)$ is a concave-convex mapping, i.e.,

$$\mu \to \mathbb{E}_{m \sim \mu} \mathcal{D}_{\mathrm{KL}}\Big(p_\theta(\cdot|m, \mathbf{y}_{<t})\big\|q(\cdot|\mathbf{y}_{<t})\Big)$$

is concave and,

$$q \to \mathbb{E}_{m \sim \mu} \mathcal{D}_{\mathrm{KL}}\Big(p_\theta(\cdot|m, \mathbf{y}_{<t})\big\|q(\cdot|\mathbf{y}_{<t})\Big)$$

is convex for every $(\mu, q)$, respectively. Then, by classical min-max theorem von Neumann (1928), we have that (15) holds.

Finally, it remains to show that:

$$\min_{q \in \mathcal{P}(\Omega)} \mathbb{E}_{m \sim \mu} \mathcal{D}_{\mathrm{KL}}\Big(p_\theta(\cdot|m, \mathbf{y}_{<t})\big\|q(\cdot|\mathbf{y}_{<t})\Big) = \mathcal{I}(\mathbb{M}; Y_t|\mathbf{y}_{<t}), \tag{16}$$

for any random variable $\mathbb{M}$ distributed according to the probability distribution $\mu \in \mathcal{P}(\mathcal{M})$ and each distribution $p_\theta(y_t|m, \mathbf{y}_{<t})$.

We begin by showing that:

$$\mathbb{E}_{m \sim \mu} \mathcal{D}_{\mathrm{KL}}\Big(p_\theta(\cdot|m, \mathbf{y}_{<t})\big\|q(\cdot|\mathbf{y}_{<t})\Big) \geq \mathcal{I}(\mathbb{M}; Y_t|\mathbf{y}_{<t})$$

for all distributions $q(\cdot|\mathbf{y}_{<t})$ and $p_\theta(y_t|m, \mathbf{y}_{<t})$. To this end, we consider the following identities:

$$\begin{aligned}
\mathbb{E}_{m \sim \mu} \mathcal{D}_{\mathrm{KL}}\Big(p_\theta(\cdot|m, \mathbf{y}_{<t})\big\|q(\cdot|\mathbf{y}_{<t})\Big) &= \mathbb{E}_{m \sim \mu} \mathcal{D}_{\mathrm{KL}}\Big(p_\theta(\cdot|m, \mathbf{y}_{<t})\big\|p_\theta(\cdot|\mathbf{y}_{<t})\Big) \\
&\quad + \mathcal{D}_{\mathrm{KL}}\Big(p_\theta(\cdot|\mathbf{y}_{<t})\big\|q(\cdot|\mathbf{y}_{<t})\Big) \\
&= \mathcal{I}(\mathbb{M}; Y_t|\mathbf{y}_{<t}) + \mathcal{D}_{\mathrm{KL}}\Big(p_\theta(\cdot|\mathbf{y}_{<t})\big\|q(\cdot|\mathbf{y}_{<t})\Big) \\
&\geq \mathcal{I}(\mathbb{M}; Y_t|\mathbf{y}_{<t}), \tag{17}
\end{aligned}$$

where $p_\theta(\cdot|\mathbf{y}_{<t})$ denotes the marginal distribution of $p_\theta(\cdot|m, \mathbf{y}_{<t})$ w.r.t. $\mu$ and the last inequality follows since the KL divergence is non-negative. Finally, it is easy to check that by selecting:

$$q^\star(y_t|\mathbf{y}_{<t}) = \mathbb{E}_{m \sim \mu}\big[p_\theta(y_t|m, \mathbf{y}_{<t})\big] \tag{18}$$

the lower bound in (17) is achieved:

$$\min_{q \in \mathcal{P}(\Omega)} \mathbb{E}_{m \sim \mu} \mathcal{D}_{\mathrm{KL}}\Big(p_\theta(\cdot|m, \mathbf{y}_{<t})\big\|q(\cdot|\mathbf{y}_{<t})\Big) = \mathbb{E}_{m \sim \mu} \mathcal{D}_{\mathrm{KL}}\Big(p_\theta(\cdot|m, \mathbf{y}_{<t})\big\|q^\star(\cdot|\mathbf{y}_{<t})\Big), \tag{19}$$

for every $\mu \in \mathcal{P}(\mathcal{M})$, which proves the identity in expression (16).

The claim in (6) follows by taking the maximum overall probability measures $\mu \in \mathcal{P}(\mathcal{M})$ at both sides of (16), and combining the resulting identity with expressions (15) and (12). The mixture in (8) follows from expression (18) which is a necessary condition to solve the min-max problem. $\qquad\square$

## B  BLAHUT−ARIMOTO ALGORITHM

### B.1  ALGORITHM

Our channel can be specified using two discrete random variables $(\mathbb{M}, Y_t)$ with alphabets $(\mathcal{M}, \Omega)$ and probability distributions $\mu$ and $p_\theta(y_t|m, \mathbf{y}_{<t})$, respectively, conditioned on $\mathbf{y}_{<t}$. The problem to be solved is the maximization of the mutual information:

$$\Gamma(\mathbf{y}_{<t}) \triangleq \max_{\mu \in \mathcal{P}(\mathcal{M})} \mathcal{I}_\theta\big(\mathbb{M}; Y_t|\mathbf{y}_{<t}\big). \tag{20}$$

Now if we denote the cardinality $|\mathbb{M}| = M$, $|\Omega| = N$, then $p_\theta(y_t|m, \mathbf{y}_{<t})$ is an $M \times N$ matrix, which we denote the $i$-th row, $j$-th column entry by $w_{ij}$. For the case of channel capacity, the algorithm was introduced in Arimoto (1972); Blahut (1972) to solve (20). They both found the following expression for the capacity of a discrete channel with channel law $w_{ij}$:

$$\Gamma(\mathbf{y}_{<t}) = \max_\mu \max_Q \sum_{i=1}^M \sum_{j=1}^N \mu_i w_{ij} \log \left( \frac{q_{ji}}{\mu_i} \right),$$

where $\mu$ and $Q$ are maximized over the following requirements:

- $\mu \triangleq (\mu_1, \ldots, \mu_M)$ is a probability distribution on $\mathcal{M}$. That is, $\sum_{i=1}^M \mu_i = 1$.
- $Q = (q_{ji})$ is an $N \times M$ matrix that behaves like a transition matrix from $\Omega$ to $\mathcal{M}$ with respect to the channel law. That is, for all $1 \le i \le M$, $1 \le j \le N$:

$$q_{ji} \ge 0, \quad q_{ji} = 0 \Leftrightarrow w_{ij} = 0,$$

and every row sums up to 1: $\sum_{i=1}^M q_{ji} = 1$.

Then, upon initializing a probability measure $\mu^0 = (\mu_1^0, \mu_2^0, \ldots, \mu_M^0)$ on $\mathcal{M}$, we can generate a sequence $(\mu^0, Q^0, \mu^1, Q^1, \ldots)$ iteratively as follows:

$$(q_{ji}^t) = \frac{\mu_i^t w_{ij}}{\displaystyle\sum_{k=1}^M \mu_k^t w_{kj}}, \tag{21}$$

and

$$\mu_k^{t+1} = \frac{\displaystyle\prod_{j=1}^N (q_{jk}^t)^{w_{kj}}}{\displaystyle\sum_{i=1}^M \prod_{j=1}^N (q_{ji}^t)^{w_{ij}}}, \tag{22}$$

for $t = 0, 1, 2, \ldots$.

Then, using the theory of optimization, specifically coordinate descent, it has been shown that the sequence indeed converges to the required maximum. That is,

$$\lim_{t \to \infty} \sum_{i=1}^M \sum_{j=1}^N \mu_i^t w_{ij} \log \left( \frac{q_{ji}^t}{\mu_i^t} \right) = \Gamma(\mathbf{y}_{<t}).$$

So given a channel law $p_\theta(y_t|m, \mathbf{y}_{<t})$, the (20) can be numerically estimated up to arbitrary precision.

### B.2 COMPUTATIONAL COMPLEXITY

The computational complexity of the Blahut-Arimoto algorithm can be characterized as follows:

- **Number of iterations.** The algorithm typically converges linearly, so the number of iterations required, denoted as $T$, is proportional to the desired accuracy of the solution.
- **Operations per iteration.** Each iteration involves updating the probability measures in (21) and (22), and evaluating the mutual information, which requires matrix manipulations. Let $M$ and $N$ be the cardinalities of the input and output alphabets, respectively. Each iteration involves operations overall input-output pairs, requiring $\mathcal{O}(M \times N)$ operations.

Combining these, the overall computational complexity of the Blahut-Arimoto algorithm is $\mathcal{O}(T \times n \times m)$, reflecting its dependence on the sizes of $M$ (number of LLMs in the considered family) and $N$ (the vocabulary), and the number of iterations needed for convergence, which depends intrinsically on the underlying distributions.

## C    BINOCULAR SCORES

The binoculars score $B(\mathbf{w})$ for an input sequence $\mathbf{w} = \langle w_0, w_1, \ldots \rangle$ is defined by

$$
B(\mathbf{w}) \triangleq \frac{\sum_{t=1}^{T} \sum_{y_t \in \Omega} \mathbb{1}[y_t = w_t] \mathcal{L}_\theta(y_t | m, \mathbf{w}_{<t})}{\sum_{t=1}^{T} \sum_{y_t \in \Omega} p_\theta(y_t | m', \mathbf{w}_{<t}) \mathcal{L}_\theta(y_t | m, \mathbf{w}_{<t})}, \tag{23}
$$

where the choices of LLMs $p_\theta(\cdot | m, \mathbf{w}_{<t}) \in \mathcal{P}_\mathcal{M}(\mathcal{Y})$ and $p_\theta(\cdot | m', \mathbf{w}_{<t}) \in \mathcal{P}_\mathcal{M}(\mathcal{Y})$ are critical for performance and have to be optimized empirically. Indeed, this represents the main weakness of this score, since in practice the best choice for the best pair of LLMs $(m, m')$ may not be distribution-free.

## D    COMPLEXITY IMPROVEMENTS

Our algorithm currently processes each text in approximately 10 seconds on NVIDIA 32G V100 GPUs. Runtime optimization is an area that should be improved in future work. Below, we outline limitations of our system and propose potential improvements : In MOSAIC, the texts are processed one-by-one by the LLMs. Each model is loaded onto a separate GPU, and the logits are moved to a central device for performing operations such as Blahut-Arimoto, perplexity, and cross-entropy calculations, after which the final score is computed. This setup has several inefficiencies. For instance, transferring logits to a central device introduces a significant bottleneck. Additionally, while calculations are performed on one GPU, the remaining ones remain idle, resulting in suboptimal use of resources.

A more efficient method would involve computing the logits for all texts in parallel, storing them across different GPUs, and performing subsequent calculations concurrently. An even more streamlined solution would involve loading all models onto a single GPU using quantized or distilled versions, thus eliminating the need to transfer logits across devices.

While these optimizations are promising, they have not been implemented in this work, as we focus on the algorithmic methodology rather than runtime efficiency.

## E    ROBUSTNESS RESULTS

Table 5 is a more complete version of Table 3, where we also include the details of all individual detectors based on just one model.

Table 6 displays TPR @ 5% FPR obtained when running our system on the RAID test dataset. It can be seen that adding repetition penalty (the w/ r_p columns) makes our results drop significantly. Note that, as no labels are provided, this is a different metric from the other tables.

Figure 2 represents how Arimoto weights evolve when looking at another language. As the Tower-Basemodels have been trained on more multilingual data, they have more importance when looking at Bulgarian text.

## F    INCLUDING THE GENERATOR IN THE ENSEMBLE

In this section, we report in Table 8 the TPR @5% FPR corresponding to the AUROC scores in Table 4. We also display in Figure 4 the changes in Blahut-Arimoto weights when simulating the effect of nucleus-sampling on the logits computed by Llama-2-7b.

## G    INCREASING THE SIZE OF THE ENSEMBLE

In this section, we add a Phi-3 model and look at the difference in results, tables 9 and Arimoto weights, Figure 5, as well as when adding a Unigram model on Figure 6.

Table 5: Artificial text detection performance of detectors built with a fixed set of 4 models. Detection may involve running 1, 2, or 4 models. AUROC scores.

| | RAID | Binoculars | | | Ghostbuster | | | M4 (multilingual) | | | | Scho. | Avg |
|---|---|---|---|---|---|---|---|---|---|---|---|---|---|
| | | Pubmed | CNN | CCnews | Reddit | Reuter | Essay | Ara | Bul | Ger | Rus | | |
| **1 model** | | | | | | | | | | | | | |
| PPL based detectors with ... | | | | | | | | | | | | | |
| Tower-7b | 0.709 | 0.798 | 0.557 | 0.582 | 0.973 | 0.961 | 0.990 | 0.882 | 0.839 | 0.639 | 0.523 | 0.503 | 0.746 |
| Tower-13b | 0.705 | 0.823 | 0.561 | 0.578 | 0.976 | 0.964 | 0.991 | 0.879 | 0.837 | 0.598 | 0.514 | 0.522 | 0.746 |
| Llama-2-7b | 0.723 | 0.786 | 0.544 | 0.556 | 0.977 | 0.971 | 0.991 | 0.888 | 0.896 | 0.597 | 0.626 | 0.520 | 0.756 |
| Llama-2-7b-chat | 0.769 | 0.817 | 0.598 | 0.617 | 0.989 | 0.994 | 0.994 | 0.918 | 0.932 | 0.693 | 0.695 | 0.520 | 0.795 |
| DetectGPT-based detectors with ... | | | | | | | | | | | | | |
| Tower-7b | 0.481 | 0.448 | 0.551 | 0.473 | 0.635 | 0.578 | 0.888 | 0.579 | 0.667 | 0.299 | 0.630 | 0.476 | 0.559 |
| Tower-13b | 0.449 | 0.489 | 0.555 | 0.489 | 0.616 | 0.555 | 0.892 | 0.578 | 0.642 | 0.267 | 0.623 | 0.480 | 0.553 |
| Llama-2-7b | 0.498 | 0.430 | 0.530 | 0.451 | 0.674 | 0.650 | 0.925 | 0.606 | 0.690 | 0.278 | 0.656 | 0.477 | 0.572 |
| Llama-2-7b-chat | 0.598 | 0.450 | 0.559 | 0.476 | 0.852 | 0.810 | 0.949 | 0.735 | 0.758 | 0.391 | 0.684 | 0.485 | 0.646 |
| FastDetectGPT-based detectors with ... | | | | | | | | | | | | | |
| Tower-7b | 0.825 | 0.997 | 0.997 | 0.965 | 0.914 | 0.834 | 0.849 | 0.973 | 0.979 | 0.896 | 0.781 | 0.531 | 0.878 |
| Tower-13b | 0.834 | 0.999 | 0.995 | 0.975 | 0.878 | 0.886 | 0.818 | 0.985 | 0.988 | 0.832 | 0.816 | 0.517 | 0.877 |
| Llama-2-7b | 0.810 | 0.993 | 0.994 | 0.959 | 0.647 | 0.616 | 0.463 | 0.954 | 0.990 | 0.893 | 0.879 | 0.511 | 0.809 |
| Llama-2-7b-chat | 0.744 | 0.955 | 0.877 | 0.896 | 0.324 | 0.650 | 0.127 | 0.870 | 0.772 | 0.644 | 0.591 | 0.395 | 0.654 |
| **2 models** | | | | | | | | | | | | | |
| Bino-best | 0.803 | 0.989 | 0.994 | 0.973 | 0.677 | 0.663 | 0.481 | 0.897 | 0.959 | 0.860 | 0.492 | 0.381 | 0.764 |
| **4 models** | | | | | | | | | | | | | |
| avg PPL | 0.730 | 0.808 | 0.566 | 0.584 | 0.980 | 0.980 | 0.992 | 0.897 | 0.887 | 0.638 | 0.598 | 0.516 | 0.765 |
| $q^\star$ (log-probs) | 0.746 | 0.807 | 0.566 | 0.579 | 0.985 | 0.984 | 0.994 | 0.893 | 0.901 | 0.640 | 0.592 | 0.509 | 0.766 |
| MOSAIC-4 (avg) | 0.850 | 0.992 | 0.993 | 0.971 | 0.946 | 0.971 | 0.911 | 0.909 | 0.974 | 0.890 | 0.737 | 0.421 | 0.880 |
| MOSAIC-4 (unif) | 0.844 | 0.992 | 0.995 | 0.975 | 0.920 | 0.951 | 0.876 | 0.909 | 0.974 | 0.893 | 0.745 | 0.416 | 0.874 |

Table 6: MOSAIC performance under different generator configurations on the RAID test dataset. Scores are TPR@5%FPR.

| | Greedy | Greedy w/o r_p | Greedy w/ r_p | Sampling | Sampling w/o r_p | Sampling w/ r_p | Repetition Penalty | No Repetition Penalty | All |
|---|---|---|---|---|---|---|---|---|---|
| MOSAIC-4 | 0.902 | 0.952 | 0.810 | 0.603 | 0.785 | 0.269 | 0.540 | 0.868 | 0.752 |
| MOSAIC-5 | 0.884 | 0.927 | 0.806 | 0.606 | 0.799 | 0.252 | 0.529 | 0.863 | 0.745 |

Table 7: MOSAIC under different adversarial attacks. Scores are TPR@5%FPR.

| | All | White Space | Upper Lower | Synonym | Miss-spelling | Para-phrase | Number Shuffling | Add Paragraphs | Homo-glyph | Article Deletion | Change Spelling | Zero Width Space |
|---|---|---|---|---|---|---|---|---|---|---|---|---|
| MOSAIC-4 | 0.693 | 0.675 | 0.686 | 0.285 | 0.725 | 0.719 | 0.713 | 0.745 | 0.866 | 0.708 | 0.729 | 0.714 |
| MOSAIC-5 | 0.694 | 0.670 | 0.665 | 0.227 | 0.717 | 0.703 | 0.697 | 0.733 | 0.902 | 0.695 | 0.722 | 0.855 |

Table 8: Identification results with varying generators for the Binoculars dataset: the original version (Llama-13b and Falcon), and our regenerated corpus with Llama-2-7B (Llama) and Mistral-7B (Mist.) (TPR@5%FPR).

| | Pubmed | | | CNN | | | CC_news | | | Avg. |
|---|---|---|---|---|---|---|---|---|---|---|
| | Orig. | Llama | Mist. | Orig. | Llama | Mist. | Orig. | Llama | Mist. | |
| **PPL TowerBase-7b** | 0.318 | 0.216 | 0.476 | 0.051 | 0.998 | 0.106 | 0.204 | 0.026 | 0.178 | 0.286 |
| **PPL TowerBase-13b** | 0.365 | 0.099 | 0.480 | 0.066 | 0.999 | 0.109 | 0.182 | 0.016 | 0.174 | 0.277 |
| **PPL Llama-2-7b-chat** | 0.353 | 0.297 | 0.546 | 0.060 | 1.000 | 0.183 | 0.230 | 0.027 | 0.215 | 0.323 |
| **PPL Llama-2-7b** | 0.293 | 0.279 | 0.500 | 0.049 | 1.000 | 0.173 | 0.187 | 0.033 | 0.227 | 0.305 |
| $q^\star$ **log-probs** | 0.330 | 0.206 | 0.533 | 0.059 | 0.999 | 0.191 | 0.195 | 0.024 | 0.245 | 0.309 |
| **MOSAIC-4** (avg) | 0.963 | 0.615 | 0.814 | 0.971 | 1.000 | 0.898 | 0.868 | 0.552 | 0.730 | 0.823 |

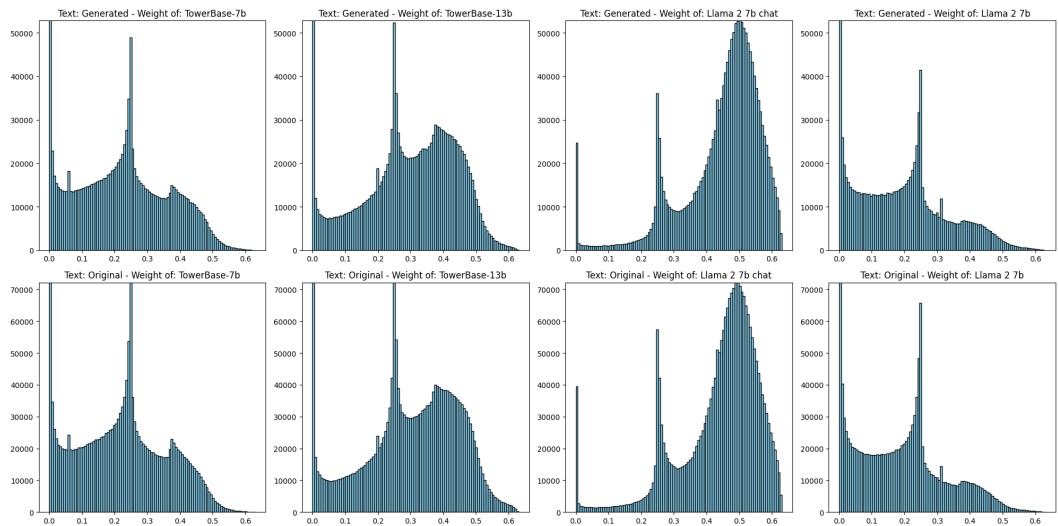

(a) Blahut–Arimoto weights for Llama-2-7b generations in English (CC_news dataset).

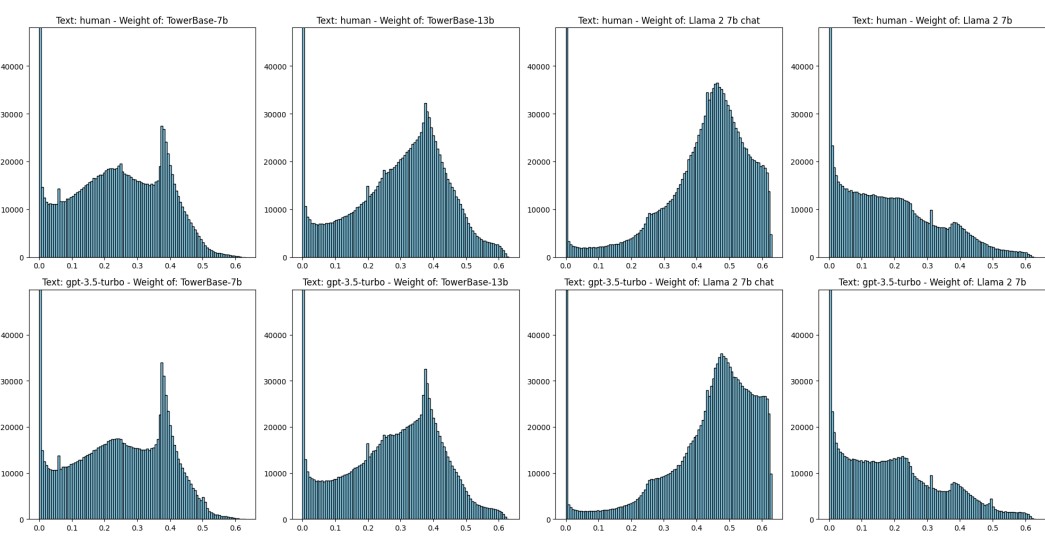

(b) Blahut–Arimoto weights for ChatGPT generations in Bulgarian.

Figure 2: Comparison of Blahut–Arimoto weights between English (CC_news) and Bulgarian (M4).

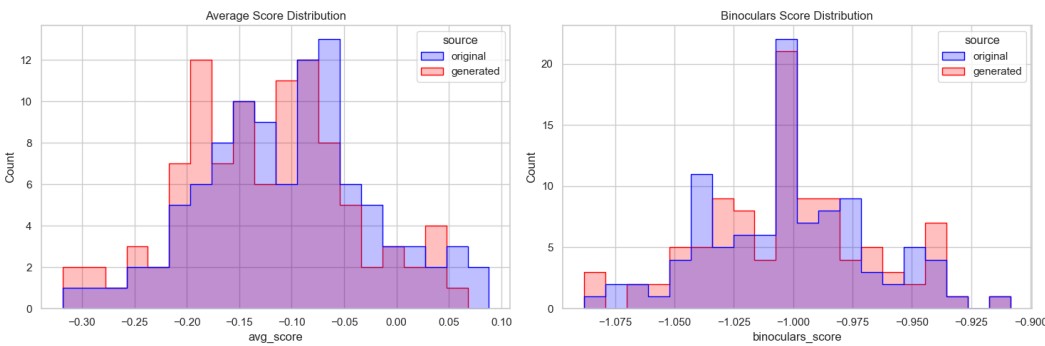

Figure 3: Scores obtained on the Academic dataset, MOSAIC on the left and Binoculars on the right, for both scoring methods, generated and original texts are indistinguishable

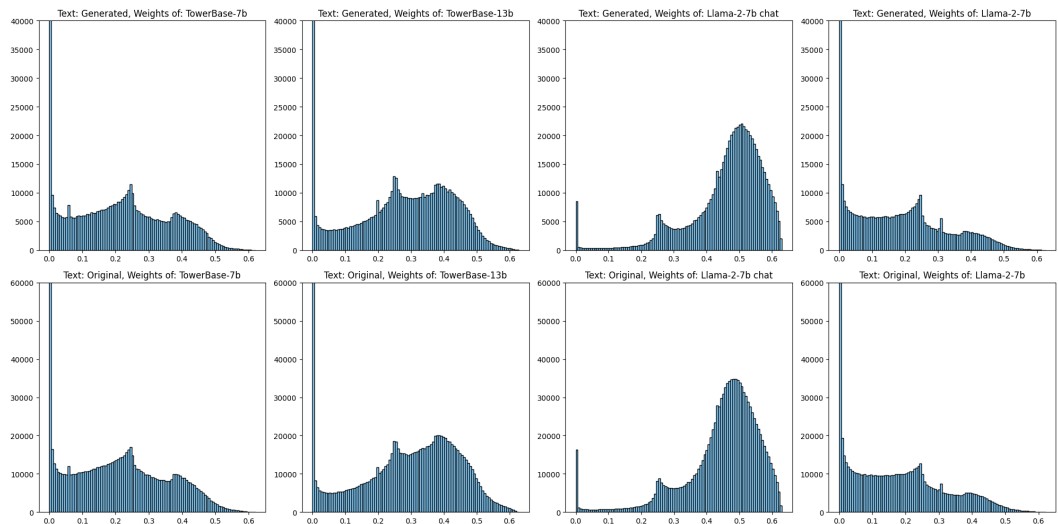

(a) Blahut–Arimoto weights for Pubmed regenerated with Llama-2-7b.

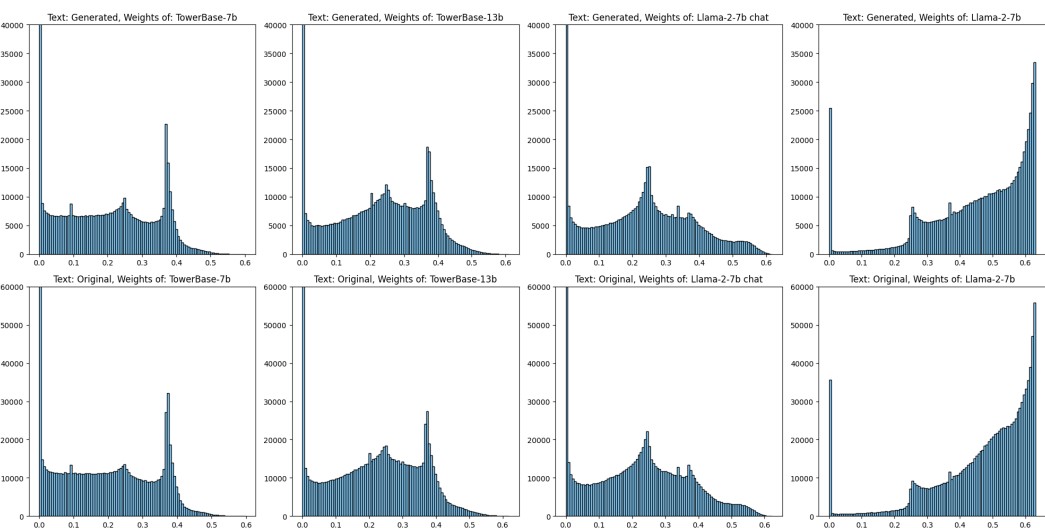

(b) Blahut–Arimoto weights for Pubmed regenerated with Llama-2-7b. In the detector, the logits for Llama-2-7b are modified to simulate the effect of nucleus sampling.

Figure 4: Comparison of Blahut–Arimoto weights with and without sampling on Llama-2-7b logits when looking at text generated with the same parameters.

Table 9: AUROC Scores on Ghostbuster and Binoculars datasets.

| Method | Reddit | Reuter | Essay | Pubmed | CNN | CC_news | Avg. |
|---|---|---|---|---|---|---|---|
| **PPL TowerBase-7b** | 0.973 | 0.961 | 0.990 | 0.709 | 0.798 | 0.557 | 0.831 |
| **PPL TowerBase-13b** | 0.976 | 0.964 | 0.991 | 0.705 | 0.823 | 0.561 | 0.837 |
| **PPL Llama-2-7b-chat** | 0.989 | 0.994 | 0.994 | 0.769 | 0.817 | 0.598 | 0.860 |
| **PPL Llama-2-7b** | 0.977 | 0.971 | 0.991 | 0.723 | 0.786 | 0.544 | 0.832 |
| **PPL Phi-3-mini-4k-instruct** | 0.993 | 0.990 | 0.995 | 0.586 | 0.997 | 0.495 | 0.843 |
| $q^\star$ **log-probs** | 0.985 | 0.984 | 0.994 | 0.807 | 0.566 | 0.579 | 0.819 |
| $q^\star$ **(+phi) log-probs** | 0.992 | 0.990 | 0.995 | 0.632 | 1.000 | 0.561 | 0.862 |
| **MOSAIC-4 (avg)** | 0.946 | 0.971 | 0.911 | 0.992 | 0.993 | 0.971 | 0.964 |
| **MOSAIC-5 (+phi, avg)** | 0.975 | 0.986 | 0.966 | 0.992 | 0.988 | 0.950 | 0.976 |
| **MOSAIC-5 (+1gram, avg)** | - | - | - | 0.986 | 0.960 | 0.898 | |

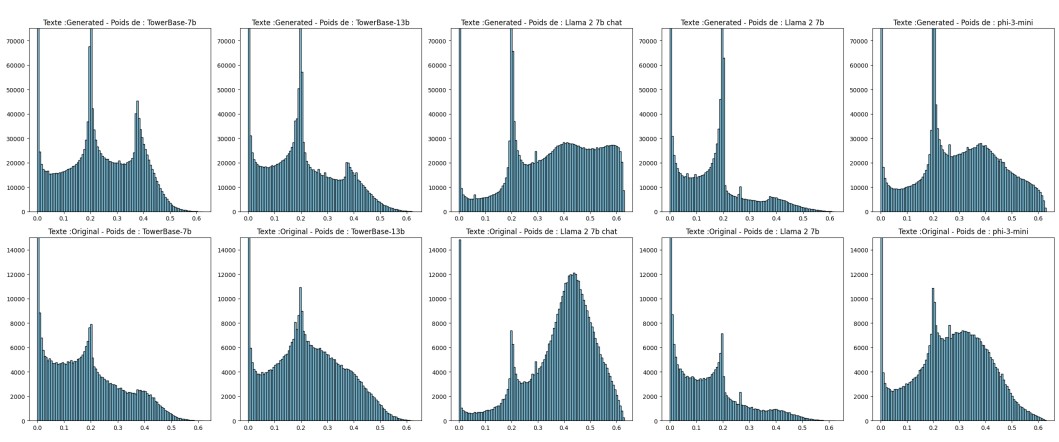

(a) Weights for the essay dataset when phi is in the ensemble.

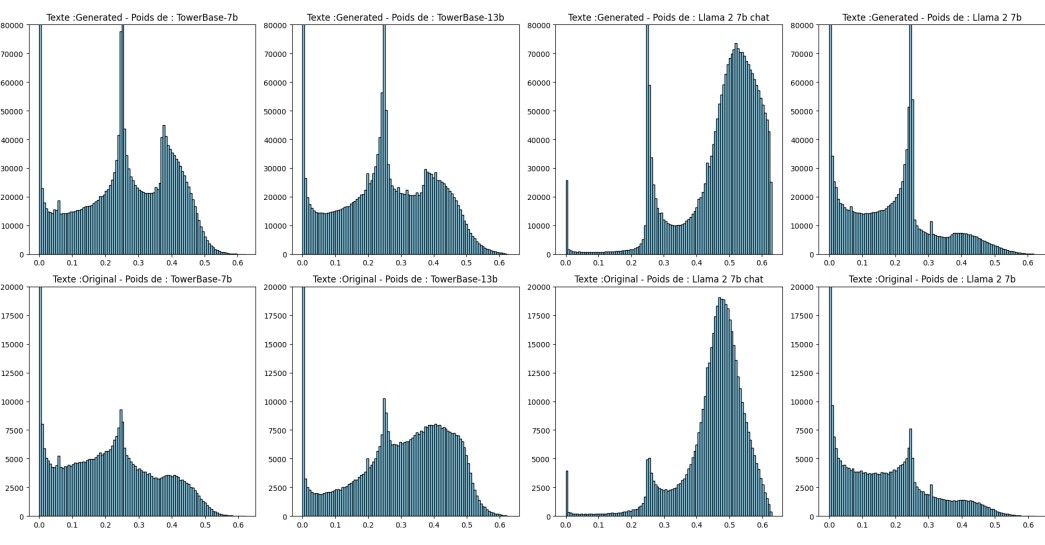

(b) Weights for the essay dataset for the ensemble without phi.

Figure 5: Comparison of Blahut–Arimoto weights between English (CC_news) and Bulgarian (M4).

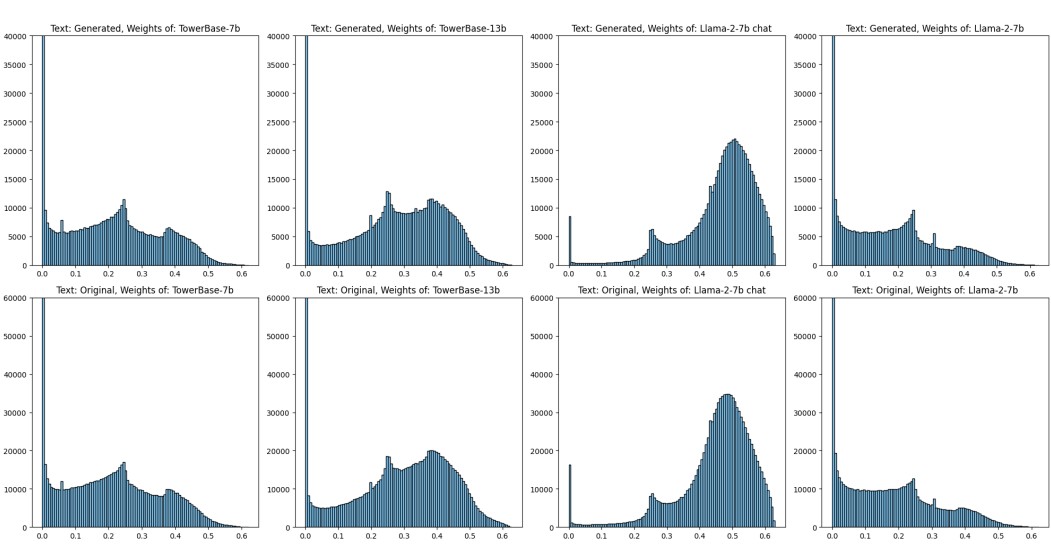

(a) Weights for the pubmed dataset without the unigram model.

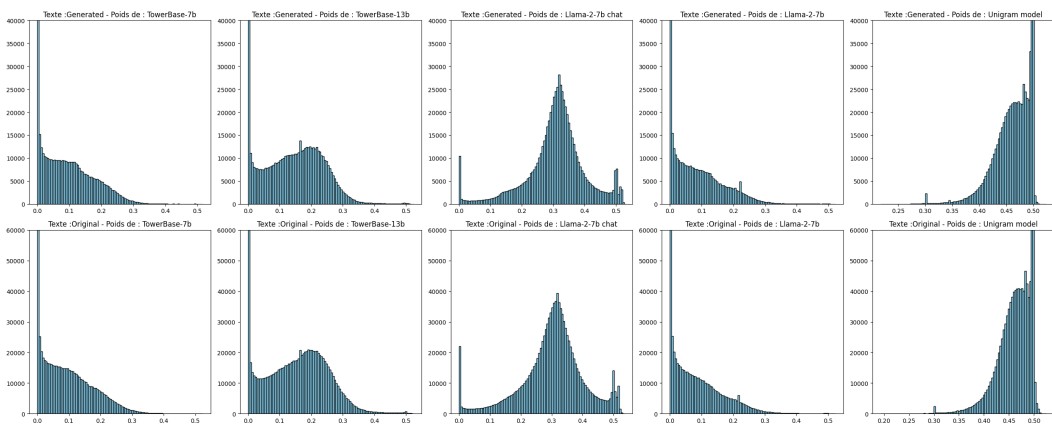

(b) Weights for the pubmed dataset when the unigram is in the ensemble.

Figure 6: Comparison of Blahut–Arimoto weights when adding the unigram to our ensemble.

