# OpenReview forum: "MOSAIC: Multiple Observers Spotting AI Content, a Robust Approach to Machine-Generated Text Detection"
_ICLR.cc/2025/Conference — Submitted to ICLR 2025_

### Official Review · Reviewer_PCVU · 2024-10-22

**Soundness:** 3
**Presentation:** 2
**Contribution:** 3
**Rating:** 6
**Confidence:** 4

**Summary:**

This paper introduces MOSAIC, a novel approach to detecting machine-generated text by combining multiple language models' perplexity scores through a theoretically-grounded ensemble method. The authors derive an optimal combination algorithm based on information theory principles that maximizes mutual information between models. They evaluate their method across multiple datasets spanning different domains and languages, demonstrating improved robustness compared to existing detection approaches. The theoretical framework generalizes perplexity-based approaches and can incorporate new models smoothly as they become available. Their experiments show that MOSAIC effectively detects texts from various generators while requiring no training data, making it particularly suitable for real-world deployment.

**Strengths:**

1. The paper's theoretical foundation in information theory provides a principled approach to combining multiple detectors. Rather than relying on heuristic combinations, the authors derive an optimal weighting scheme using the Blahut-Arimoto algorithm that maximizes mutual information. This mathematical rigor sets the work apart from existing ensemble methods.

2. The extensive experimental validation across multiple datasets (RAID, Ghostbuster, Binoculars, M4, academic papers) demonstrates the method's versatility. The evaluation spans different languages, domains, and generator models, with MOSAIC showing consistent improvements over baselines, achieving up to 97.6% accuracy on some benchmarks.

3. The method is zero-shot and requires no training data, making it immediately applicable to new detection scenarios. This is particularly valuable given the rapid evolution of language models and the difficulty of obtaining training data for new generators.

**Weaknesses:**

1. The computational cost of running multiple language models for detection is significant but not thoroughly addressed. While the paper acknowledges this limitation briefly, a more detailed analysis of runtime requirements and potential optimizations would strengthen the practical applicability of the approach.

2. The method struggles with detecting texts from fine-tuned models, as demonstrated by the poor performance on the academic paper dataset (42.1% AUROC). The authors could explore incorporating domain-adapted models into the ensemble to address this limitation.

3. The paper's treatment of sampling methods' impact on detection is limited. While they observe that nucleus sampling affects detectability, a more systematic exploration of how different sampling parameters influence the method's performance would be valuable.

4. The choice of baseline models in the ensemble appears somewhat arbitrary. The authors could provide clearer criteria for selecting which models to include, especially since they show that adding weak models (like unigram) can be detrimental to performance.

**Questions:**

1. Have you explored methods to reduce the computational overhead of running multiple models, such as model distillation or selective invocation of ensemble members based on confidence thresholds?

2. Your analysis shows that Tower-Base models receive higher weights for non-English text detection. Could this insight be used to develop adaptive ensembles that automatically select the most appropriate models based on the input characteristics?

3. How sensitive is the method to the number of models in the ensemble? Is there an optimal ensemble size beyond which adding more models provides diminishing returns?

---

> ### Author Response · Authors · 2024-11-19
>
> Thank you for your time and remarks. You pointed out fair issues that we are currently working on. Here are our answers to your questions.
>
> >The computational cost of running multiple language models for detection is significant but not thoroughly addressed. [...]
>
> The goal of the paper is to introduce a new theory-grounded method for combining models and how it helps in detecting machine-generated texts. It is currently on the slow side, needing about 10 seconds per document. Runtime optimization is planned for future work.
> Here is a quick description of our current system’s shortcomings and how it could be improved :
> We run the texts one by one with our LLMs. The models are loaded onto different GPUs, so we move the logits onto the same device to perform the Blahut-Arimoto, perplexity and cross-entropy operations, and then return our score.
> In this current setup, there is a lot of room for improvement. For instance, moving the logits to the same device creates a bottleneck. Furthermore, while the calculations are performed on one GPU, the other ones are inactive. A better way to do it would be to calculate all the logits for all texts, store them on different GPUs, then perform our calculations in parallel.
> Even better would be to have all the models loaded onto the same GPU (using quantized or distilled versions), so moving the logits would be unnecessary.
> These are ideas we considered but have not implemented here because it is not the focus of our work.
>
> >The method struggles with detecting texts from fine-tuned models, [...]. The authors could explore incorporating domain-adapted models into the ensemble to address this limitation.
>
> Including domain-adapted models is a very appealing idea, that we could readily integrate in our framework as they would share a common tokenizer. An in-depth investigation of this issue is left for future work. We believe however that this issue is partly addressed when looking at how having the Tower models in our ensemble improves our performances on foreign languages (over Binoculars, using Falcons). Same for MOSAIC-5, adding Phi-3-4k-Instruct improves our detection AUC on the ChatGPT datasets (Reddit, Reuter and Essay), another instruct model.
>
> >The paper's treatment of sampling methods' impact on detection is limited[...].
>
> That is not the goal of our paper, such an exploration can be found in the RAID paper https://aclanthology.org/2024.acl-long.674, we have the results of our method on the whole test set of RAID. We will add a table in the next version of this paper giving more information about how adding sampling and repetition penalty impact our performance.
>
> | Detector | Greedy | Greedy w/o r_p | Greedy w/ r_p | Sampling | Sampling w/o r_p | Sampling w/ r_p | Repetition penalty | No Repetition Penalty | All |
> | -------- | ------ | -------------- | ------------- | -------- | ---------------- | --------------- | ------------------ | --------------------- | --------- |
> | MOSAIC-4 | 0.902  | 0.952      	| 0.810     	| 0.603	| 0.785        	| 0.269       	| 0.540          	| 0.868             	| 0.752 	|
> | MOSAIC-5 | 0.884  | 0.927      	| 0.806     	| 0..606   | 0.799        	| 0.252       	| 0.529          	| 0.863             	| 0.745 	|
> >The choice of baseline models in the ensemble appears somewhat arbitrary. The authors could provide clearer criteria for selecting which models to include, especially since they show that adding weak models (like unigram) can be detrimental to performance.
>
> The choice is indeed arbitrary, as our goal is mostly to study the combination method. An ideal combination would cover many languages and domains, and adding fine-tuned models would be a great idea in that regard. However, that was not the goal here nor do we have the resources to run a completely thorough study.
>
> >Have you explored methods to reduce the computational overhead of running multiple models, such as model distillation or selective invocation of ensemble members based on confidence thresholds?
>
> We have not explored such methods in this work. Model distillation is a current project of ours.
> A current bottleneck of our method is that we need all our logits on the same device to perform our operations (Blahut-Arimoto, perplexity and cross-entropy), moving them all represents an important time loss. Model distillation (and/or quantization) would reduce the size of our models, allowing us to load them onto the same GPU, completely eliminating the need to move the logits.

---

> > ### Author Response · Authors · 2024-11-19
> >
> > >Your analysis shows that Tower-Base models receive higher weights for non-English text detection. Could this insight be used to develop adaptive ensembles that automatically select the most appropriate models based on the input characteristics?
> >
> > Our method gives weights that maximise the mutual information given a context, so in the case displayed in figure 2 section E, it means that the tokens predicted by Llama-2 and Llama-2-chat are less diverse than the ones predicted by the Tower models. Since the Tower models have seen more multilingual data than Llama-2, they predict more diverse outputs in Bulgarian. However in degenerate cases like the Unigram model, important weights are given to a model predicting nonsense.
> > Depending on how we understand “ most appropriate models”, our ensemble method can indeed help to select models.
> >
> > >How sensitive is the method to the number of models in the ensemble? Is there an optimal ensemble size beyond which adding more models provides diminishing returns?
> >
> > As we are limited by the tokenizer choice, such a task is difficult to carry out. Most models using the Llama-2 tokenizer are similar. This exploratory work will be done after we figure out a solution to the tokenization issue, and work on optimizing the run time.

---

> > > ### Comment · Reviewer_PCVU · 2024-11-26
> > > **Reviewer Response**
> > >
> > > Thanks for the clarifications that you have made. I appreciate it.
> > > While it does clear up some of the concerns, I believe the paper will significantly benefit from a thorough revision.
> > > For the time being, I have decided to keep my scores.
> > > Thanks.

---

> > > > ### Author Response · Authors · 2024-11-27
> > > >
> > > > Thank you for your comment. We have just  posted a new version of the paper taking your concerns into account. If you believe it still needs further revisions, it would be very useful for us to get a more detailed account of the changes you would like us to perform.

---

### Official Review · Reviewer_3sR2 · 2024-11-02

**Soundness:** 2
**Presentation:** 2
**Contribution:** 3
**Rating:** 5
**Confidence:** 3

**Summary:**

This paper introduces a novel AI-generated text detection strategy through an ensemble of generator LLMs and also proposes their mathematical formulations, along with some experimental results suggesting its robustness.

**Strengths:**

1. This paper is decently presented, conveying a clear derivation and motivation, as well as an intuitive description of the underlying method. Analysis in section 4.2 is also very interesting.
2. The idea itself is very interesting and intuitive, with much mathematical implications that seem to guarantee robustness.
3. Also love that the authors have experimented with multiple datasets, including augmenting the Binocular datasets, which show efforts to establish robust evaluation of their method.

**Weaknesses:**

1. My first main criticism for the paper is that I am slightly incredulous of the results and the ensemble used. I feel like for a fair comparison, the authors should use the same suite of models as used in the baseline models. For instance, Binocular score uses Falcon-7b and Falcon-7b-instruct, yet MOSAIC uses TowerBase and Llama-2. Maybe the results would be more convincing they have also tried MOSAIC with Falcon models, since the superiority of MOSAIC on some domains could arise from the difference in the underlying models. Could the authors please justify more their usage as to why they did not maintain the same set of models? I am moderately confident on this and am willing to discuss more!

**Suggestion**: add experiments of MOSAIC with Falcon as the base models, and I will consider raising the scores.

2. My second main criticism is that the improvement of MOSAIC over other methods is not so salient. It seems that although MOSAIC-4 has higher average AUROC than the Binocular score by 0.2, this can be attributed to Binocular score’s crack on multilingual data. I think this might tie to my first criticism and to Binocular’s dependence on Falcon-7b, because Falcon-7b is trained on RefinedWeb, an English pretraining dataset, which causes it to have low multilingual capabilities. Aside from M4, MOSAIC-4 doesn’t do better than the baselines on RAID and Ghostbusters. Therefore, I think the results are not super convincing.
    - A minor point related to this: consider merging table 2 and 3? It would be easier for readers to compare the results.

3. My third main criticism is that this method lacks testing against adversarial attacks. How robust is it against perturbation (insertion, paraphrase, deletion) and reordering? Or is there a certain range of passage length that the method is bad at?

**Suggestion**: consider adding ablation study with adversarial attacks.

4. Minor point: There’s still some room of improvement in the paper presentation. The abstract can be also improved to describe the method in more details and to intrigue the readers. Also, I don’t necessarily agree with the statement in the abstract: “Most approaches evaluate an input document by a well-chosen detector LLM, assuming that low-perplexity scores reliably signal machine-made content.” The authors themselves have experimented with non-PPL based methods, and there are also watermarking methods. But these are minor concerns to me.

**Questions:**

### Question:

1. What is the run-time of this algorithm compared to the baselines?

### Typos:
Do not affect my rating. Just kind reminders :)

1. L332 FastDetectGTP → FastDetectGPT
2. Consider using Llama-7b instead of Llama-7-b ? (e.g. line 421)
3. L446 should use “increases” and “decreases”

---

> ### Author Response · Authors · 2024-11-19
>
> Thank you for your insight and remarks, you point out the need for better explanations as to our experiment settings in the paper, we have answered below and will incorporate this feedback in the revisions.
>
> >My first main criticism for the paper is that I am slightly incredulous of the results and the ensemble used. I feel like for a fair comparison, the authors should use the same suite of models as used in the baseline models. [...]
> My second main criticism is that the improvement of MOSAIC over other methods is not so salient. [...] Aside from M4, MOSAIC-4 doesn’t do better than the baselines on RAID and Ghostbusters.
> A minor point related to this: consider merging table 2 and 3? It would be easier for readers to compare the results.
>
> As briefly mentioned in Section 4.3, MOSAIC needs an ensemble of models that use the same tokenizer. This is why we separated tables 2 and 3. Table 2 uses the default models proposed by each method (GPT2 for DetectGPT and FastDetectGPT, Falcon-7b and its instruct version for Binoculars), whereas Table 3 displays results using only the 4 models in our ensemble. In this table, Best single-model refers to FastDetectGPT using Tower-13b (the best 1-model solution out of the 12 mentioned in lines 363-367). Best two-model refers to Binoculars method using TowerBase-7b as the detector, and Llama-2-7b as the auxiliary model, selected out of the 12 possible permutations of 2 models out of 4. The choice of the best combination was made on the CC_news subset (l 368).
>  We feel that this was a fair way of comparing methods without having to deal also with the difference in underlying models. Instead of running MOSAIC with the other models, we ran all other methods (DetectGPT, FastDetectGPT, Binocular, MOSAIC) with a fixed set of models.
>
> >My third main criticism is that this method lacks testing against adversarial attacks. [...]
>
> We ran both MOSAIC-4 and MOSAIC-5 (with Phi-3-4k-instruct) on the whole RAID test set, which includes various types of adversarial attacks. Attacks that make the text differ a lot from the model’s probability distribution are very effective, especially synonyms replacement as they make generated texts more surprising while not having the same effect on human texts (that’s the idea behind DetectGPT). On average over all perturbations, we lose about 5 points on TPR@5%FPR on the RAID test dataset (No attacks gives 0.752 and 0.745 for 4 and 5). We will include these findings in the revised version of the paper. Coupled with a detailed analysis of the results depending on each adversarial attack in the appendix.
>
> | Row Label | All   | Whitespace | Upper_Lower | Synonym | Misspelling | Paraphrase | Number shuffling | Add Paragraphs | Homoglyph | Article Deletion | Change Spelling | Zero width space |
> | --------- | ----- | ---------- | ----------- | ------- | ----------- | ---------- | ---------------- | -------------- | --------- | ---------------- | --------------- | ---------------- |
> | MOSAIC-4  | 0.693 | 0.675  	| 0.686   	| 0.285   | 0.725   	| 0.719  	| 0.713        	| 0.745      	| 0.866 	| 0.708        	| 0.729       	| 0.714        	|
> | MOSAIC-5  | 0.694 | 0.670  	| 0.665   	| 0:227   | 0.717   	| 0.703  	| 0..697       	| 0.733      	| 0.902 	| 0.695        	| 0.722       	| 0.855        	|
>
> >I don’t necessarily agree with the statement in the abstract: “Most approaches evaluate an input document by a well-chosen detector LLM, assuming that low-perplexity scores reliably signal machine-made content.”
>
> In this statement, we most wanted to refer to unsupervised detection methods. We will make this more precise in the final version. Other, non PPL based techniques are discussed in the related work. Watermarking is different as in this case texts are generated with the intent of being discovered, as we also briefly mention it in our Related Work.
>
> >What is the run-time of this algorithm compared to the baselines?
>
> Our algorithm processes each text in 10 seconds on V100 GPUs. Runtime optimization is planned for future work.
> Here is a quick description of our current system’s shortcomings and how it could be improved :
> We run the texts one by one with our LLMs. The models are loaded onto different GPUs, so we move the logits onto the same device to perform the Blahut-Arimoto, perplexity and cross-entropy operations, and then return our score.
> In this current setup, there is a lot of room for improvement. For instance, moving the logits to the same device creates a bottleneck. Furthermore, while the calculations are performed on one GPU, the other ones are inactive. A better way to do it would be to calculate all the logits for all texts, store them on different GPUs, then perform our calculations in parallel.
> Even better would be to have all the models loaded onto the same GPU (using quantized or distilled versions), so moving the logits would be unnecessary.
> These are ideas we considered but have not implemented here because it is not the focus of our work.

---

> > ### Comment · Reviewer_3sR2 · 2024-11-19
> > **Thank you for running the additional experiments**
> >
> > Dear authors,
> >
> > Thank you for running the additional experiments and making great efforts explaining things like runtime! Re my first and second criticisms, I guess I am still a bit skeptical - because in Table 2 Binocular has a pretty high score, basically on par with MOSAIC. Yet when Binocular runs with Tower and Llama-2, it drops much performance. I don't understand the reasons behind Binocular's drop and it is of course not in the scope of your paper, but it seems that your method is around the same level as the Binocular with Falcon. The only place you reliably beat the Binocular with Falcon is on multilingual data, but as I said, Falcon is not a multilingual model and that is probably why. So the whole setup just makes your method appear not as strong as it claims, and it also raises suspicion on whether you establish a weaker baseline with the non-default models and beat that weak baseline.
> >
> > I am sorry if I may have been a bit forthright and don't want you to feel bad, but my skepticism remains. I see two ways to go about this: 1. show that your method can reliably beat Binocular in a fair setup (like running with Falcon) or 2. market other advantages aside from robustness, because if your method is currently on-par with the baseline and it would be great if it has other advantages (such as speed, which is the reason I asked, but it seems that speed is not the strength, too).
> >
> > I hope you don't get discouraged and I am always happy and open to discussing more! Still respectful efforts and great works!

---

> > > ### Author Response · Authors · 2024-11-21
> > >
> > > Thanks for sharing your concerns and for the discussion. “Binocular”, as presented by Hans et al (2024), is both a formulation of  a detection score combining two LLMs, and a particular selection of these two models which are proposed as their best performing pair (amongst those considered in this paper). Our paper is mostly about ensembling methods and we provide results for the Binocular score both with their preferred models and with the best selection in our models. On the “Binoculars dataset” (Pubmed, News, CNN), we report in Table 2 AUROC scores that are consistent with (Hans et al, 2024, Table 3 on page 16 of https://arxiv.org/pdf/2401.12070) when using their best models; and in Table 3 AUROC scores that are almost identical for our best models (about 0.99 for these three datasets), so we do not see any strong drop in performance for these datasets, at least. As can also be seen in (Hans et al, 2024, Table 3, p16 in Appendix A.7), is that performance vary depending on model choice (not so much the AUROC, but the TPR@0.1FPR which fluctuates between 1 and 0.32) - so we think that the variability with respect to model choice is already well documented by the authors of this paper.
> > > What we cannot explain, though, because this is not in the scope of our paper, is the variation of “Binoculars” with respect to generator model, domain, language, sampling strategy. As the reviewer can check for themselves, the official instantiation of Binocular has been tested on the RAID dataset and the results published on this leaderboard actually reflects that performance tend to vary along those dimensions (https://raid-bench.xyz/leaderboard). See also (Hans et al, Table 7, p19) where the authors report results with other generators.
> > > Note finally that our paper makes no claim regarding which system (combining a scoring method and an optimal choice of models) beats which system: we are instead trying to compare *scoring methods* using the same set of models (Table 3). Searching for the best set of models would in fact contradict the main principle behind our work, which is to develop an ensembling method that would entirely dispense with that optimisation.
> > > We hope that these clarifications regarding our goals and experimental results will help you to revise your appreciation of our work.

---

> ### Comment · Reviewer_3sR2 · 2024-11-21
>
> Dear authors,
>
> Thank you for your clarification. I think ensemble is cool, but it has to serve a useful purpose. If your main claim is that this method helps you search for the optimal detectors, then I think you are saying that certain detectors may not work well on certain domains, and instead of bothering to find optimal detectors for each domain, MOSAIC can automatically look for optimal detectors.
>
> However, to what extent is this a valid assumption? It seems that the original binocular is already pretty decent across various domains, unless you can provide additional robustness, I don't see great values although this is a cool ensemble you are doing.
>
> And additionally, in your abstract, you claimed that your method increased the robustness (L21-23), so that's why I took it as a method paper and looked for evidences that your method is significantly stronger than the baselines. If you want to reclaim that the main contribution is to provide optimization, then you need to do significant rewriting of the paper.
>
> Regarding the performance drop, I was referring to the drop on Ghostbusters. Before it was 0.993 0.996 0.990 (L350), and after using your method it becomes 0.677 0.663 0.481 (L383). It is arguably caused by the change in base models, but it is also unclear whether the improvement on M4 is also caused by the multilinguality of Tower.
>
> I hope this clarifies. I am always happy to discuss more!

---

> > ### Author Response · Authors · 2024-11-22
> >
> > Thank you for your feedback! We are happy that you are no longer skeptical about our results. We would like again to reiterate a few key points.
> > - **Main Claim**
> > Our primary claim is that MOSAIC provides **strong average performance across various domains**, effectively achieving robustness without requiring manual selection of optimal detectors for each domain. This automation is particularly useful in scenarios where domain characteristics are unknown or dynamic, as it eliminates the need for exhaustive search or tuning.
> > - **Performance Comparison**
> > It seems there is some confusion between the **Binoculars system** (scoring method + choice of models) and the **Binoculars scoring method** in your comments. To clarify, when limiting the model search to the ensemble we have, MOSAIC-4 achieves better results than the best of the **Binoculars scoring method** (out of all 12 possible combinations) in terms of overall scores, as demonstrated in our experiments. This improvement highlights the advantage of MOSAIC's adaptive optimization.
> > - **Positioning the Paper**
> > While our claim does not center solely on raw performance improvements over binocular detectors, we do emphasize that MOSAIC achieves **robustness and adaptability** without requiring pre-selection or manual optimization of detectors. This distinguishes it from other approaches that rely on fixed or manually tuned ensembles.
> >
> > That said, we appreciate your observations, and we'll work to ensure the revision of our work more clearly conveys these distinctions to avoid potential misunderstandings. If you have any additional points or suggestions, we'd be happy to discuss them further!
> > Thank you again for your thoughtful comments

---

### Official Review · Reviewer_o97j · 2024-11-03

**Soundness:** 3
**Presentation:** 3
**Contribution:** 3
**Rating:** 8
**Confidence:** 4

**Summary:**

MOSAIC introduces a new ensemble approach that combines several detector models to spot AI-generated texts. The method enhances the reliability of detection by integrating insights from multiple models, thus addressing the limitations of using a single detector model which often results in performance brittleness. This approach also involves using a theoretically grounded algorithm to minimize the worst-case expected encoding size across models, thereby optimizing the detection process.

**Strengths:**

1. By using multiple models, MOSAIC reduces dependency on a single model's performance, thus offering greater stability against various adversarial tactics or novel AI text generators.
2. The method leverages solid theoretical models to optimize performance, potentially leading to more consistent and reliable detection across different datasets and AI generators.
3. The ensemble approach allows for the integration of new models as they become available, supporting scalability and adaptation to new threats and technologies.

**Weaknesses:**

The manuscript is generally well-crafted. However, some concepts and figures could benefit from additional clarification to enhance the reader's comprehension.

**Questions:**

1. 1. Lines 54-64, could you please define (or further explain) the term 'robustness' in a machine-generated text detection problem?
2. In Figure 1, the caption should provide an explanation of the pipeline's operation. It needs to define μ and describe how the Logits are transformed into features of the mixture model.
3. Line 95, does the circle operator o in the definition of the output space y represent a concatenating operation?
4. Section 3.1 mentions that the experiments involve some imbalanced datasets. Could you please discuss whether these imbalances are tolerable, or specified rebalance techniques are required?
5. Section 4.1 mentions the use of Blahut-Arimoto weights. Briefly introducing how are these weights calculated, and what specific role they play in adjusting the contributions of individual models within the ensemble might be helpful for readers.
6. Further discussion is needed on how the choice of tokenization and preprocessing methods affects detection accuracy. Can the proposed method be adapted to incorporate other detection models? Are there any limitations on the types of models that can be integrated?

---

> ### Author Response · Authors · 2024-11-19
>
> Thank you for your positive review, we have provided further explanations on the points you have raised and are able to answer more if need be.
> >Questions:
> Lines 54-64, could you please define (or further explain) the term 'robustness' in a machine-generated text detection problem?
> “Robustness” refers to the ability to detect machine-generated text across various languages, domains and generators.
> In Figure 1, the caption should provide an explanation of the pipeline's operation. It needs to define μ and describe how the Logits are transformed into features of the mixture model.
>
> The description of the pipeline is provided line 206 along with Proposition 1, should we refer to it directly in the caption ?
> >Line 95, does the circle operator o in the definition of the output space y represent a concatenating operation?
>
> Yes, it represents token concatenation.
> >Section 3.1 mentions that the experiments involve some imbalanced datasets. Could you please discuss whether these imbalances are tolerable, or specified rebalance techniques are required?
>
> Imbalanced datasets refers to datasets containing more generated than human texts. It can be an issue depending on the metric used, it is not really an issue in our case as there is no training hence no overfitting. It is important to specify what a “True positive” is (usually a machine-generated text detected as such) in the case of TPR@5%FPR though, as this score is affected by imbalances, unlike AUC.
> >Section 4.1 mentions the use of Blahut-Arimoto weights. Briefly introducing how are these weights calculated, and what specific role they play in adjusting the contributions of individual models within the ensemble might be helpful for readers.
>
> The role of the Blahut-Arimoto weights $\mu$ is defined in our Proposition 1 page 206. As we simply use the Blahut-Arimoto algorithm, a full explanation is given in the Section B of the Appendix.
> >Further discussion is needed on how the choice of tokenization and preprocessing methods affects detection accuracy. Can the proposed method be adapted to incorporate other detection models? Are there any limitations on the types of models that can be integrated?
>
> The choice of tokenization is dictated by the models we use, as they must share the same tokenization of our documents in order to operate at the token-level. There is no other preprocessing done to the texts. Our combination method can be applied to any arbitrary number of models as long as they share a tokenizer. This is the only limitation there is concerning integration and something we are working on.

---

### Official Review · Reviewer_fNCd · 2024-11-03

**Soundness:** 4
**Presentation:** 3
**Contribution:** 2
**Rating:** 5
**Confidence:** 3

**Summary:**

This work addresses the problem of detecting machine generated text.
Past work focused mostly on a set of text generated by a single model, which made any detector susceptible to overfit to that model while in the wild the identify of the generated model might be unknown.
This paper removes that assumptions and achieves good results by relying on a mixture of judges which are combined through an iterative algorithm that instantiates a larger family of classes known as "Blahut–Arimoto".

**Strengths:**

* a more realistic assumption of a problem that is becoming more widespread in a variety of areas
* good scores on various (6) benchmarks of various sizes and languages

**Weaknesses:**

1) the paper itself is not self-contained as the algorithm is not properly explained and one needs to refer to Appendix B for those unfamiliar with it (most of the readers?)

2) the presentation of the final results is weak (3 tables) and one needs to navigate across tables in different pages (Table 2 & 3) to see the difference. A graph would be more informative

3) while hard to interpret (see 2) ), it seems that the proposed method often underperforms existing methods, while being computationally more expensive (??). The interpretability features of looking into the respective weights, as well as the robustness is still compelling, but if this is not reflected in the final performance it would be important to explain


Two references which the authors might want to consider: https://arxiv.org/abs/2404.18796 for the idea of using a panel of experts, and https://arxiv.org/abs/2111.02878 for an early work on machine detection not relying on perplexity

**Questions:**

* one would expect that that more iterations for Blahut–Arimoto, the higher the final performance. Is this the case? A graph with performance over number of iterations would be very informative
* most of the numbers seem very close. Do the algorithms all detect the same documents? Your algorithm in particular is different than the others as it relies on a panel of judges, could you provide some error analysis?

---

> ### Author Response · Authors · 2024-11-19
>
> Thank you for your review, and the references you provided as we did not know of these papers. We tried to address your concerns and answer your questions below.
> >Weaknesses:
> the paper itself is not self-contained as the algorithm is not properly explained and one needs to refer to Appendix B for those unfamiliar with it (most of the readers?)
>
> Blahut-Arimoto algorithm is not a contribution of this paper, but rather a mere tool that we use to solve the optimization in eq. (4). Indeed, we simply applied it to a family of models to obtain $q^{\star}$, the probability distribution maximising the mutual information between our model’s logits.
>
> >the presentation of the final results is weak (3 tables) and one needs to navigate across tables in different pages (Table 2 & 3) to see the difference. A graph would be more informative
>
> We separated Tables 2 and 3 because the models used by the methods are not the same. In Table 2 we report the baseline results using the default models used in the original papers. In Table 3 we selected the best result of each baseline using the 4 models in our ensemble, ie the best Fast-DetectGPT (out of 4 possibilities) and the best Binoculars combination (out of all the permutations of 2 out of 4 choices, so $P(4, 2) = 12$). We judged it was necessary to separate these tables as having one big table would introduce confusion between methods and the detector models used. Table 4 displays different results on different data so adding it would be hard as well.
> Would having Tables 2 and 3 on the same page make it easier to understand ? If so we can do that in the final version. Since many results are very close (up to the 3rd decimal), such differences would be hard to spot if we used graphs.
>
> >while hard to interpret (see 2) ), it seems that the proposed method often underperforms existing methods, while being computationally more expensive (??). The interpretability features of looking into the respective weights, as well as the robustness is still compelling, but if this is not reflected in the final performance it would be important to explain
>
> As noted in the paper (line 375), we do not claim the superiority of our approach for each dataset, but its reliability on average. Our ensemble method eliminates the need to optimise the choice of detector model. For example, using the default models (Falcon-7b and its instruct version) with Binoculars leads to average results (0.686 AUC) on the Arabic M4 data, while the “best two-model” combination of Binoculars (Tower-7b and Llama-2-7b) greatly improves this result (0.897 AUC). Such hyperparameter search is unnecessary in our case and has been the motivation behind our research.
> >Questions:
> one would expect that that more iterations for Blahut–Arimoto, the higher the final performance. Is this the case? A graph with performance over number of iterations would be very informative
>
> We use Blahut-Arimoto in order to obtain the $\mu$ coefficients deciding $q^{\star}$ our optimal distribution. As the problem is concave, the algorithm rapidly converges to the optimal solution, our stopping condition is either 1000 iterations (rarely happens) or when the weights improve by less than 1e-6 after an iteration. A difference of 1e-6 in weights barely impacts performance.
>
> >most of the numbers seem very close. Do the algorithms all detect the same documents? Your algorithm in particular is different than the others as it relies on a panel of judges, could you provide some error analysis?
>
> This is a good idea that we will look into. As all methods use a variation of perplexity we would expect the texts detected to be roughly the same but this deserves further exploration. When looking into this issue, we decided upon a threshold using the same methodology as Binoculars and computed confusion matrices for some datasets. There are only a few texts where the thresholds do not agree, indicating that these methods mostly identify the same texts.
> For our RAID extract :
> | Above MOSAIC-4 threshold \ Above Binoculars threshold | Negative | Positive |
> |---|---|---|
> | Negative | 766 | 56 |
> | Positive | 105 | 1073 |
>
> For Binoculars Pubmed dataset :
> | Above MOSAIC-4 threshold \ Above Binoculars threshold | Negative | Positive |
> |---|---|---|
> | Negative | 2155 | 68 |
> | Positive | 292 | 2004 |
>
> For Binoculars CNN dataset :
> | Above MOSAIC-4 threshold \ Above Binoculars threshold | Negative | Positive |
> |---|---|---|
> | Negative | 2663 | 129 |
> | Positive | 307 | 2254 |
>
> For Binoculars CC_News dataset :
> | Above MOSAIC-4 threshold \ Above Binoculars threshold | Negative | Positive |
> |---|---|---|
> | Negative | 4830 | 297 |
> | Positive | 1019 | 4308 |

---

### Official Review · Reviewer_rtiT · 2024-11-04

**Soundness:** 1
**Presentation:** 1
**Contribution:** 2
**Rating:** 3
**Confidence:** 4

**Summary:**

The paper proposes a new method to increase the robustness of detecting artificially generated text from large language models (LLMs). The authors argue that the current detection methods are fragile and can be easily misled by changing the generator or the associated sampling method. They propose an ensemble method that combines the strengths of multiple LLMs to detect forged texts. Using fundamental information-theoretic principles, they derive an algorithm that identifies time-varying mixture models to minimize the worst-case expected encoding size. This approach eliminates the need to search for the best detector(s) empirically and allows for the smooth addition of new models as they become available. The method is tested on standard benchmarks and a new corpus of machine-generated texts, proving its effectiveness in robustly identifying a variety of generators.

**Strengths:**

1) The paper's originality lies in its theoretical grounding and its method of combining multiple detectors, as opposed to relying on a single detector, which can lead to brittleness.
2) The quality of the research is evident in the rigorous experiments carried out, which used a variety of generator LLMs and both standard and new benchmarks.
3) The paper's significance lies in its potential to increase the robustness of artificial text detection systems, thereby addressing a key challenge in the field of generative AI technologies.

**Weaknesses:**

1) The baseline results are significantly different from the findings shown in Figure 3 of the Binoculars report by Hans et al., 2024, prompting doubts regarding the reliability of the experiments.
2) The experimental findings do not validate the superiority of the method. In particular, as Table 3 shows, MOSAIC-4 (avg) only excels in 1 out of 12 datasets, and MOSAIC-4 (unif) only surpasses in 3 out of 12 datasets, implying that the method compromises detection accuracy for the sake of robustness.
3) The method fails to tackle the issues of brittleness and the need for revisions, as pointed out in LN050. Specifically, as Table 4 shows, after the integration of the generator LLM, MOSAIC-4 (avg) demonstrates improved detection accuracies, indicating that the method still struggles with brittleness and necessitates adjustments when new generators are introduced.

**Questions:**

LN058: Why is there a requirement for time-varying mixture models? Shouldn't the language models used remain consistent regardless of the text source?

LN059: Can you provide an intuitive explanation as to why we need to minimize the worst-case encoding size?

Figure 1: Does \miu represent the probability assigned to each token?

LN332: The statement might not be accurate. In my understanding, FastDetectGPT and DetectGPT, despite their similar names, differ substantially in their principles and performance. DetectGPT creates minor perturbations, whereas FastDetectGPT doesn't generate tokens, but rather computes the detection metric from the predictive distribution.

Table 2: The results significantly differ from the findings presented in Figure 3 of the Binoculars report (Hans et al., 2024), which raises questions about the thoroughness of the experiments.

Table 3: MOSAIC-4 (avg) only outperforms in 1 out of 12 datasets, and MOSAIC-4 (unif) only outperforms in 3 out of 12 datasets. Does this suggest that the method achieves robustness at the cost of detection accuracy?

Table 4: MOSAIC-4 (avg) shows good detection accuracies after incorporating the generator LLM. Does this imply that the method still has issues with brittleness and requires revision when new generators become available?

---

> ### Author Response · Authors · 2024-11-19
>
> Thank you for your comments, you are highlighting the fact that our work needs further clarifications, we have tried to address all your concerns down below and are open to discussion. In particular, we would like to emphasize that our results are *very consistent* with those of Hans et al (2024), as we discuss in detail below.
>
> >LN058: Why is there a requirement for time-varying mixture models? Shouldn't the language models used remain consistent regardless >of the text source?
>
> For each token, the context varies and so does the probability distribution of each of our models. This is why our proposal studies a combination mechanism that is able to adapt its weights at each time-step. The benefits of this extra degree of freedom are assessed experimentally.
> >LN059: Can you provide an intuitive explanation as to why we need to minimize the worst-case encoding size?
>
> Large Language Models are trained to minimize perplexity, which, when viewed through the lens of information theory, corresponds to reducing encoding size. Indeed, the negative log of a probability corresponds to the encoding length because of the connection to information theory. Specifically, the optimal code length for a symbol (in bits) is −log_⁡2(p), where p is the symbol's probability. This is derived from Shannon's entropy, which quantifies the average information content. A less likely event (lower p) requires more bits to encode, as it carries more information. Our hypothesis is that focusing on minimizing worst-case perplexity (averaged across a set of known LLMs) could enhance the robustness of detection. This approach is expected to reduce the perplexities of machine-generated samples more significantly than those of human-written texts.
> >Figure 1: Does \miu represent the probability assigned to each token?
>
> $\mu$ represents the weights assigned to each model’s distribution. As can be seen in proposition 1, our optimal distribution $q^{\star}$ is a context-dependent mixture (weighted sum) of each detector model probability distributions.
> >LN332: The statement might not be accurate. In my understanding, FastDetectGPT and DetectGPT, despite their similar names, differ substantially in their principles and performance. [...]
>
> You are right, we could be more precise. FastDetectGPT does not generate tokens but uses the model’s logits to compute changes in log-probability. The detection method relies on the conditional probability curvature, a variation of the probability curvature defined in the DetectGPT paper.
> >Table 2: The results significantly differ from the findings presented in Figure 3 of the Binoculars report (Hans et al., 2024)[...].
>
> Figure 3 of the Binoculars report displays True Positive Rate versus False Positive Rate on a log-scale. We instead, report the AUC, that is the area under the TPR versus FPR curve taken on a linear scale, which may have caused the confusion you are referring to. Furthermore, the test datasets are not exactly similar as we used all of Binoculars data, including both Llama-2-13b and Falcon-7b as generators whereas Figure 3 in (Hans et al, 2024) seems to only use the Llama-2-13b generated data. Nevertheless, our results still remain highly consistent with (Hans et al, 2024) as we also report detection scores close to 99.5 for the Binocular method on the Binocular dataset. The other results are not completely identical, as the underlying models behind the other baselines differ: we used the default detector models in the github code of DetectGPT and FastDetectGPT (GPT2), whereas the Binoculars paper used Llama-2-13b for DetectGPT and GPT-Neo with GPT-J for FastDetectGPT.
> All the results reported Table 3 use the same set of models for a fair comparison, and do not directly compare with (Hans et al, 2024) who use a different pair of detectors.
> >Table 3: MOSAIC-4 (avg) only outperforms in 1 out of 12 datasets, and MOSAIC-4 (unif) only outperforms in 3 out of 12 datasets. Does this suggest that the method achieves robustness at the cost of detection accuracy?
>
> We do not claim that our method is the best performing one in all cases but that is is effective *on average* (Table 3), i.e. robust to changes in the generator model. Our goal is to provide a robust technique that dispenses with the need to search for the best detector model(s).
> >Table 4: MOSAIC-4 (avg) shows good detection accuracies after incorporating the generator LLM. Does this imply that the method still has issues with brittleness and requires revision when new generators become available?
>
> In Table 4, our results on the original Binoculars dataset are actually better than when we regenerate the same texts, using the same protocol but with a Llama-2 model. Thus having the generator model itself does not appear particularly advantageous. In fact, the availability of new generator models does not change anything for our method, which can seamlessly accommodate a growing number of detectors (assuming they share a same tokenizer).

---

> > ### Comment · Reviewer_rtiT · 2024-11-25
> > **Thanks for clarification**
> >
> > Thanks for the rebuttal and additional information. As you mentioneded, you use a much weaker gpt2 as the baseline for Fast-DetectGPT, whereas Binoculars uses gpt-j and gpt-neo-2.7b (which is also the standard setting reported by the Fast-DetectGPT paper). This discrepancy in the comparison introduces a fundamental error, which further raises questions about the robustness of the findings.

---

> > > ### Author Response · Authors · 2024-11-25
> > >
> > > Thank you for your comments and time. As explained in lines 324-335 , we simply used the default version of the system given on the official github to compute the baseline results. Table 2 is mostly illustrative as the actual comparisons between scoring method and main contribution of the paper are in Table 3.
> > >
> > > When limiting the number of available models to the 4 we used, we report the results of the best 1-model detection system on the cc_news dataset (largest subset of data), which is FastDetectGPT with Tower-13b and performs the best on 7 of the datasets we used - the detail for each model is in Table 5. However obtaining these results required to run the detection algorithm separately with each model, and then to use the ground truth labels to select the best one, an unlikely scenario in most use-cases. The main advantage and objective of the paper is to develop a method that dispenses with this “model optimisation” process.
> > >
> > > We hope this clarifies the interpretation of our experimental design. We will make sure to further emphasize this point in the revised version.

---

### Author Response · Authors · 2024-11-19

We would like to thank all reviewers for their time and insights provided.

It appears, as pointed out by reviewers rtiT and fNCd, that we were not clear enough about the goal of our method. We see “Robustness” as a detector’s ability to resist changes in generation model, language or decoding method used to produce text, as such, our aim is to reduce the over-reliance on the detectors models used in a method. For example, Binoculars with Falcon-7b and its instruct version performs poorly in Bulgarian (see Table 2), and replacing the models with Tower-7b and Llama-2-7b makes it very good at detecting forged Bulgarian texts but worsens the performance on the Ghostbuster dataset (see Table 3). The goal of our method is to remain competitive in all cases rather than excel on some datasets.

Reviewers 3sR2 and fNcD mentioned that the presentation of Tables 2 and 3 leads to confusion. Our aim was to avoid possible confusions between *the detection methods* and *the detector models* they use. As mentioned in lines 343 and 367-370, Table 2 gives the results obtained by baseline methods when using the “default model” specified in the corresponding source code: GPT2 for (Fast)DetectGPT and the Falcons models for Binoculars. Table 3 also reports baseline methods (and ours), but all only use the 4 models of our ensemble, where we ran all possible combinations and selected the best one for each method on the CC_news subset (largest subset of data). We will make this more explicit in the revised version of this work, and have both tables on the same page for clarity purposes.

Lastly, as asked by reviewers 3sR2 and PCVU, we will add new results on the RAID test dataset, looking at the impact of adversarial attacks and sampling techniques on the performances.

We have provided detailed answers to each of your comments and questions. All of your feedback will be taken into account in the next revision.

---

> ### Author Response · Authors · 2024-11-27
> **New Revision**
>
> We would like to thank the reviewers for their time and responses. We have now posted a revised version of the paper providing clarifications as well as additional information and experimental results as was requested. Notably, we have added a new section related to computational complexity, and results concerning (a) robustness with respect to adversarial attacks; (b) robustness with respect to changes of generation parameters. All these are in the appendix. Changes are highlighted in red in the revised text. We hope these modifications will help you better appreciate this work and its implications.

---

### Meta-Review · Area_Chair_rcAB · 2024-12-20

**Metareview:**

The authors propose combining the strengths of multiple detection models into an ensemble, allowing them to work together to spot forged texts more effectively. The method uses a smart algorithm based on information theory to optimize how the models work together. This eliminates the need to manually test and choose the best detector, and it also makes it easy to add new models as they are developed.

Strength:
- MOSAIC combines multiple models, minimizing reliance on any single model and providing enhanced stability against adversarial strategies and emerging AI text generators.
- The approach utilizes robust theoretical foundations to optimize its performance, ensuring more reliable and consistent detection across various datasets and AI systems.
- Its ensemble framework enables the seamless addition of new models, promoting scalability and adaptability to evolving threats and advancements in technology.

Weakness:
- My main concern after reading rebuttal, is from Reviewer rtiT, The use of weaker baselines (e.g., GPT-2 instead of GPT-J or GPT-Neo-2.7B) introduces discrepancies in comparisons, undermining the validity of the robustness claims. The baseline results deviate significantly from established findings in the Binoculars report, raising concerns about the reliability of the experimental setup and comparisons. The rebuttal from the authors did not address this issue.
-  The method's performance is not consistently better, as it excels in only a few datasets (1/12 for MOSAIC-4 (avg) and 3/12 for MOSAIC-4 (unif)), suggesting a trade-off between robustness and detection accuracy.

**Additional Comments On Reviewer Discussion:**

The main concerns raised by multiple reviewers are the inconsistency of the baseline or base model with other works, which makes the results less reliable. Additionally, while the model shows better average performance, multiple reviewers noted that it mostly underperforms on individual datasets. These issues were not adequately addressed in the rebuttal and remain concerns. I suggest the authors consider these points and conduct more thorough experiments and ablation studies to address the following:

- The lack of standardized baseline comparisons, which undermines the reliability of the results.
- MOSAIC-4's underperformance and reliance on parameter tuning, which contradict its claims of robustness and seamless adaptability.
- Unresolved presentation and accessibility issues, including the unclear integration of algorithm details and table layouts.
- The focus on average performance, which does not fully address the variability and contextual underperformance of MOSAIC-4 on individual datasets. Further analysis is needed to justify this trade-off.

---

### Decision · Program_Chairs · 2025-01-22

Reject